



# 1 Assessing the accuracy of microwave radiometers and

# 2 radio acoustic sounding systems for wind energy

# 3 applications

Laura Bianco[1,2], Katja Friedrich[3], James Wilczak[2], Duane Hazen[1,2], Daniel Wolfe[1,2], Ruben
Delgado[4], Steve Oncley[5], and Julie K. Lundquist[3,6]
[1] Cooperative Institute for Research in Environmental Sciences (CIRES), Boulder, CO, USA
[2] National Oceanic and Atmospheric Administration/Earth Systems Research
Laboratory/Physical Science division, Boulder, CO, USA
[3] Department of Atmospheric and Oceanic Sciences, University of Colorado, Boulder, CO, USA
[4] University of Maryland Baltimore County, Baltimore, MA, USA
[5] National Center for Atmospheric Research, Boulder, CO, USA
[6] National Renewable Energy Laboratory, Golden, CO, USA
*Correspondence to: Laura Bianco (laura.bianco@noaa.gov)*





**Abstract.** To assess current remote-sensing capabilities for wind energy applications, a remote-
sensing system evaluation study, called XPIA (eXperimental Planetary boundary layer
Instrument Assessment), was held in the spring of 2015 at NOAA's Boulder Atmospheric
Observatory (BAO) facility. Several remote-sensing platforms were evaluated to determine their
suitability for the verification and validation processes used to test the accuracy of numerical
weather prediction models.
The evaluation of these platforms was performed with respect to well-defined reference systems:
the BAO's 300-m tower equipped at 6 levels (50, 100, 150, 200, 250, and 300m) with 12 sonic
anemometers and 6 temperature and relative humidity sensors; and approximately 60 radiosonde
launches.
In this study we first employ these reference measurements to validate temperature profiles
retrieved by two co-located microwave radiometers, as well as virtual temperature measured by
co-located wind profiling radars equipped with radio acoustic sounding systems. Results indicate
a mean absolute error in the temperature retrieved by the microwave radiometers below 1.5 °C in
the lowest 5 km of the atmosphere, and a mean absolute error in the virtual temperature
measured by the radio acoustic sounding systems below 0.8 °C in the layer of the atmosphere
covered by these measurements (up to approximately 1.6 – 2 km). We also investigated the
benefit of the vertical velocity applied to the speed of sound before computing the virtual
temperature by the radio acoustic sounding systems. We find that using this correction frequently
increases the RASS error, and that it should not be routinely applied to all data.
Water vapor density profiles measured by the MWRs were also compared with similar
measurements from the soundings, showing the capability of MWRs to follow the vertical profile
measured by the sounding, and finding a mean absolute error below 0.5 g m$^{-3}$ in the lowest 5 km



of the atmosphere. However, the relative humidity profiles measured by the microwave
radiometer lack the high-resolution details available from radiosonde profiles. An encouraging
and significant finding of this study was that the coefficient of determination between the lapse
rate measured by the microwave radiometer and the tower measurements over the tower levels
between 50 and 300 m ranged from 0.76 to 0.91, proving that these remote-sensing instruments
can provide accurate information on atmospheric stability conditions in the lower boundary
layer.





## 1. Introduction

While the increasing deployment of wind turbines increases society's reliance on
renewably-generated electricity, the need for accurate forecasts of that power production also is
growing (Marquis et al., 2011). Improving wind forecasts at hub height remains the top priority,
but challenges derive from the complexity of physical processes occurring at a wide range of
spatial and temporal scales. Fundamental to understanding and accurately forecasting these
processes is the accurate measurement of the atmospheric parameters such as wind, temperature,
and humidity in four-dimensional space. Assessing the capability and accuracy of remote-
sensing instruments to capture planetary boundary layer and flow characteristics was one goal of
the DOE- and NOAA-sponsored eXperimental Planetary boundary layer Instrumentation
Assessment (XPIA; Lundquist et al. 2016) campaign conducted in the spring 2015 at the Boulder
Atmospheric Observatory (BAO), located in Erie, Colorado (~20 km east of Boulder and ~30 km
north of Denver).
Herein, we address some of the objectives of the XPIA campaign – determining the accuracy of
temperature, water vapor density, and humidity profiles from state-of-the-art remote-sensing
instruments such as Microwave Radiometers (MWRs) and Wind Profiling Radars (WPRs)
equipped with a Radio Acoustic Sounding System (RASS), and assessing the capability of these
active and passive remote-sensing instruments to provide accurate information on atmospheric
stability conditions in the lower atmospheric boundary layer. These remote-sensing instruments
operated side-by-side during XPIA and are compared to *in-situ* observations from instruments
mounted on a 300-m meteorological tower and radiosondes.
Several studies have focused on evaluating the accuracy of temperature, water vapor density, and
humidity retrieved by MWRs (e.g., Güldner and Spänkuch 2001; Liljegren et al. 2001; Ware et





al. 2003; Cimini et al. 2011; Friedrich et al. 2012) and virtual temperature retrieved by WPRs
with RASS (May et al., 1989; Moran and Strauch, 1994; Angevine et al., 1998; Görsdorf and
Lehmann, 2000). Studies evaluating the accuracy of MWR measurements using radiosonde
observations show consistent results with differences of 1 – 2 K in temperature, <0.4 g m$^{-3}$ in
water vapor density, and < 20% in humidity for most weather conditions (Güldner and Spänkuch
2001; Liljegren et al. 2001; Ware et al. 2003; Cimini et al. 2011). Similar results were derived
from comparisons between MWR and *in-situ* tower observations with differences in temperature
ranging from 0.7 – 1.7 K (Friedrich et al. 2012). Studies evaluating the accuracy of RASS
measurements using radiosonde and *in-situ* tower observations show root mean square
differences of below 1 °C in virtual temperature (May et al., 1989; Moran and Strauch, 1994;
Angevine et al., 1998). Variations in the results were often a function of various factors such as
height above ground, season, topography, abrupt changes in the lapse rate, and regional
differences in the surrounding vegetation.
The analysis presented here builds on the results of these previous studies, but also focuses on
several unique aspects. First, in our study we provide a comprehensive assessment and
comparison of the accuracy of active (two different RASS systems operating side-by-side in
similar modes) and passive (two identical MWRs operating side-by-side in identical modes)
remote sensing instruments operated over the period of the XPIA campaign under various
meteorological conditions, including cold stable air masses, downslope wind conditions,
convective conditions, and rain and snow conditions. The accuracies of the retrievals for the two
MWR systems were compared to each other as well as to several *in-situ* radiosonde soundings
and to the tower observations. Virtual temperatures from a 915-MHz WPR with RASS and a
449-MHz WPR with RASS were also analyzed and compared to *in-situ* radiosonde soundings



and tower observations. A second important contribution of this study is to specifically
investigate and compare the ability of these active and passive remote-sensing instruments to
measure lapse rate to be used for wind energy applications. Knowing the atmospheric stability is
indubitably important for wind energy applications such as wind turbine operations, as
atmospheric turbulence and wind shear are affected by changes in atmospheric stability.
Furthermore, as found in Warthon and Lunquist (2012), and Vanderwende and Lundquist
(2012), atmospheric stability impacts both turbine power production (stable conditions improve
power performance while the opposite is true for strongly unstable conditions) and wake
characteristics (Hansen et al., 2012).
This paper is organized as follows: Section 2 summarizes the experimental design and
instrument characteristics; Sections 3 and 4 assesses the accuracy of temperature and lapse rates
derived from MWRs and RASSs, respectively; in Section 5 water vapor density and humidity
from the MWRs are compared to *in-situ* measurements. Finally, conclusions are presented in
Section 6.

**2. Experimental design, instruments and methods**
**2.1 Experimental design**
We assess temperature, water vapor density, and relative humidity accuracy from remote-sensing
instruments by comparing the observations to *in-situ* observations from radiosondes and
instruments mounted on a 300-m meteorological tower. The remote sensing instruments include
two identical 35-channel (21 in the 22-30 GHz band, and 14 in the 51-59 GHz band)
Radiometrics MWRs, one 915-MHz WPR equipped with RASS, and one 449-MHz WPR also





equipped with RASS. Figure 1 shows the instruments used in this study. A detailed description
of the instruments, methods, and their integration into the XPIA campaign can be found in
Lundquist et al. (2016). The MWRs and WPR-RASSs operated side-by-side (~2 m apart) at the
visitor center, at about 600 m southwest of the 300-m meteorological tower. Radiosondes were
launched from the visitor center in March (38 soundings), while the remaining 23 soundings
were launched in April and May from the water tank site 1000 m  to the southeast of the visitor
center (see Fig. 1 in Lundquist et al. 2016 for details).

**2.2. *In-situ* observations: Radiosonde and 300-m meteorological tower**
Radiosondes were launched during fair weather conditions at 0800 (1400), 1200 (1800), 1600
(2200), and 2000 (0200) LT (UTC) between 9 – 19 March (38 soundings), 15 and 20 – 22 April
(10 soundings), and 1 – 4 May (13 soundings) providing, among others, vertical profiles of
temperature, dewpoint temperature, and relative humidity between the surface and > 10 km
above ground level (AGL). Fourteen of these soundings were released during stable atmospheric
conditions, while the remaining forty seven were launched during unstable conditions.
Two types of sounding systems were used during the campaign: the National Center for
Atmospheric Research's Mobile GPS Advanced Upper Air Sounding System (MGAUS) was
used in March (with a 1 s temporal resolution, an accuracy of 0.25 ºC on temperature and of
1.5% on relative humidity) for launches from the visitor center, while the Vaisala MW31
DigiCORA Sounding System was used in April and May (with a 2 s temporal resolution, an
accuracy of 0. 5 ºC on temperature and of 5% on relative humidity) for launches from the water
tank site.



The 300-m meteorological tower was equipped with temperature and relative humidity sensors at
six levels (50, 100, 150, 200, 250, and 300 m) operating continuously at a temporal resolution of
1 s. Temperature was measured with an accuracy better than 0.1 K (Horst et al., 2016).

**2.3. Microwave radiometers**
Two MWRs, one operated by NOAA (referred to as the NOAA MWR hereafter) and one
operated by the University of Colorado (CU MWR), ran side-by-side with identical
configurations. Prior to the experiment, both MWRs were calibrated using an external liquid
nitrogen target and an internal ambient target (Han and Westwater 2000) and thoroughly
serviced (sensor cleaning, radome replacement etc.). Both MWRs observed at the zenith and at
an elevation angle of 15° above the ground (referred to as 15° elevation scans hereafter). The
instruments were aligned in a way that the 15° elevation scans pointed towards the north and
south, approximately parallel to the Colorado Front Range. Microwave emissions at the water
vapor (22-30 GZ) and oxygen (51-59 GHZ) absorption band together with infrared emission at
9.6-11.5 microns were used to retrieve vertical profiles of temperature ($T$), water vapor density
($WVD$) and relative humidity ($RH$) every 2-3 minutes using historic radiosondes and a regression
methods or neural network (Solheim et al. 1998a, 1998b; Ware et al. 2003). The algorithm, based
on a radiative transfer model (Rosenkranz 1998), was trained for both MWRs on a 5-year
radiosonde climatology from the Denver, Colorado, National Weather Service sounding archive,
based on radiosondes launched at the Denver International Airport, 35 km to the southeast of the
instrument site. Note that the MWR observes within an inverted cone with a 2°-3° beam width at
51-59 GHz, and 5°-6° beam width at 22-30 GHz (Ware et al. 2003). Instruments were placed
next to each other on a trailer ~3 m off the ground and at a distance of ~2 m to avoid



interference. Although the instruments use a hydrophobic radome and forced airflow over the
surface of the radome during rain, these instruments become less accurate in the presence of rain
as some water deposits on the radome. It has been observed that retrieved temperature and
humidity profiles from the 15° elevation scans provide higher accuracy during precipitation
compared to the zenith observations by minimizing the effect of liquid water and ice on the
radiometer radome (Xu et al., 2014).
The vertical resolution of the retrieved profiles ranged from 50 m between the surface and 0.5
km AGL; 100 m between 0.5 to 2 km AGL; and 250 m between 2 and 10 km AGL. Both
instruments were also equipped with a rain sensor and a surface sensor for observations of
temperature, pressure, and relative humidity. These surface observations serve as a boundary
condition for the neural network approach. Since the pressure sensor from the NOAA MWR was
broken between 5 – 27 April, the inter-comparison for the NOAA radiometer focuses solely on
observations collected between 9 March – 4 April and 28 April – 7 May 2015, while the inter-
comparison for the CU radiometer includes all dates between 9 March – 7 May 2015.

**2.4. WPR-RASSs**
Wind profiling radars are primarily used to measure the vertical profile of the horizontal wind
vector (Strauch et al., 1984; Ecklund et al., 1988). The remote measurement of virtual
temperature in the lower atmosphere is achieved with the associated RASS, co-located with the
WPR. Usually a WPR-RASS system is set up to operate in wind mode for a large fraction of
each hour and in RASS mode for the remaining small fraction. When the system is in RASS
mode, the RASS emits a longitudinal acoustic wave upward in the air that generates a local




compression and rarefaction of the ambient air. These density variations are tracked by the
Doppler radar and the speed of sound is measured. From the measurement of the speed of sound,
the virtual temperature ($T_v$) in the boundary layer can be obtained (North et al., 1973).
The 915-MHz WPR RASS settings were selected to sample the boundary layer from 120 m to
1618 m in the vertical with a 62 m resolution, while the 449-MHz WPR RASS sampled the
boundary layer from 217 m to 2001 m with a 105 m resolution.
Several factors can undermine the accuracy in $T_v$ measurements from RASSs. For example,
vertical velocity can influence the accuracy of RASS measurements (May et al., 1989; Moran
and Strauch, 1994) because the apparent speed of sound measured by the radar is equal to the
sum of the true speed of sound and the vertical air velocity. Previous studies (Moran and Strauch,
1994; Angevine and Ecklund, 1994; Görsdorf and Lehmann, 2000) have found conflicting
results on the overall accuracy of $T_v$ measurements by RASSs from correcting the speed of sound
for the vertical velocity. Görsdorf and Lehmann (2000) found that the vertical velocity correction
improves the accuracy of RASS temperature measurements only in situations when the error of
the measured vertical velocities is smaller than the magnitude of vertical velocity itself. This
situation is more likely to occur under unstable conditions in the boundary layer. In some cases,
they found that this correction can decrease the accuracy of RASS, especially in situations with
only light horizontal winds and a lower reliability of vertical wind measurements. Our systems
provided both corrected and uncorrected vertical velocity, enabling us to investigate the accuracy
of RASS measurements of $T_v$ both corrected and uncorrected for vertical air motion.





Since the volumes sampled by the MWRs and RASSs are substantially larger than those sampled
by the soundings or the tower-based measurements, vertical averaging and linear interpolation
were used to facilitate comparison. Particularly, when comparing measurements from MWRs
and RASSs to sounding observations we averaged the data of the soundings over the heights of
the MWRs and RASSs. When comparing measurements from RASSs to the tower observations
we linearly interpolated the data of the tower over the heights of the RASSs, while when
comparing measurements from the MWRs to the tower observations no spatial interpolation or
averaging was applied since MWR derived temperature levels (0, 50, 100, 150, 200, 250, 300 m)
were the same height levels as the in situ tower observations.

**3. Accuracy of the temperature profiles**
**3.1. MWRs versus sounding observations**
Differences in temperature between the two MWRs were analyzed before comparing to sounding
observations. Profiles derived from 15° elevation scans between the surface and 10 km were
compared during the time periods when both instruments were functioning (9/3/2015 – 4/4/2015
in Fig. 2a; 28/4/2015 – 7/5/2015 in Fig. 2b). Note that the off-zenith scans towards the north and
south were averaged to reduce the impact of any horizontal inhomogeneity of the atmosphere.
Although MWRs operated side-by-side with exactly the same configurations (section 2.3), mean
absolute error (MAE) between the two systems ranged between 0.7 – 0.9°C (Fig. 2). Note that
the lack of data in April was due to a malfunctioning pressure sensor of the NOAA MWR. In
general, the CU MWR observed lower temperatures ($T$) than did the NOAA MWR with the bias
between the two instruments [computed as ($T_{CU\ MWR} - T_{NOAA\ MWR}$)] ranging between -0.4 – -



0.6°C. Since the coefficient of determination, $R^2$, value was 1.00 during the inter-comparison
period, we consider the two MWRs in good agreement with each other.
For the comparison between MWRs and sounding observations, the data set was divided into
three periods in order to account for differences in the sounding systems and their locations, as
well as differences in the atmospheric conditions. The three periods of comparison consist of
March, with cooler temperatures and partially snow-covered terrain; May, with mainly warm,
convective weather; and a transition period in April with a mixture of cool, rainy and warm,
sunny weather. Differences in temperatures between the MWRs and soundings are shown for
March in Fig. 3, April in Fig. 4, and May in Fig. 5 between the surface and 5 km AGL. Scatter
plot comparisons between soundings and the radiometer observations show that MAEs in
temperature were slightly larger in March, ranging between 1.3 – 1.5 °C (Fig. 3a-b) compared to
April and May, where values ranged between 0.9 – 1.1°C (Figs. 4a, 5a-b). As previously
indicated in Fig. 2, the CU MWR underestimated the temperatures compared to the sounding
observations with a bias of -0.3 – -0.8°C in March, April and May. The NOAA MWR showed no
bias (defined as $T_{MWR}$ - $T_{Radiosonde}$) in March but overestimated temperatures in May with a bias
of 0.4°C.
Temperatures derived from the MWR zenith scans were also compared to the sounding
observations as presented in Table 1. Several studies have suggested to use off-zenith
observations at 15° to avoid temperature saturation and reduce scatter (Cimini et al., 2011;
Friedrich et al., 2012). In the present data set the zenith measurements performed better than the
averaged 15° elevation scans in terms of bias for the CU MWR in March, but not for the NOAA
MWR. Despite the higher resolution from the 15° elevation scans, values of MAE and $R^2$ are
surprisingly almost identical for the off-zenith and zenith measurements for all three periods.



Slopes are closer to one for the 15° elevation scans. We decided to base the rest of the study on
off-zenith averaged observations at 15° elevation angle because 15° elevation scans provide
higher accuracy compared to the zenith observations during precipitation (Xu et al., 2014).
MAE in temperature between the MWRs and radiosondes as a function of height, shown in Figs.
3c, 4b, 5c, indicates two different patterns in the cooler March conditions compared to a warmer
April and May. In March, MAEs were below 2°C at altitudes below 3.5 km for the CU MWR
with a continuous increase up to 2.7°C at 4.5 km AGL (Fig. 3c). The NOAA MWR showed a
similar behavior with a slightly lower MAE that the CU MWR. In April and May, however,
MAEs were below ~2°C at all levels not showing the increase in MAE that was seen in March
above 3.5 km.
Bias in temperature between the MWRs and radiosondes as a function of height (Figs. 3d, 4c,
5d) showed negative bias in a shallow layer near the surface, positive values below ~1 km (~1.5
km) for the CU (NOAA) MWR and mostly negative values above. The negative bias below 250
m is related to the surface inversions often observed at night or early morning (an example if
which is shown in Fig. 6a). The details of the inversion were consistently in error with the
MWRs too cold at the surface and too warm above a few hundred meters due to the inversion
height being displaced too high. Above 1.5 km, for some of the profiles, radiosonde temperatures
strongly differed from the MWR observations (an example if which is shown in Fig. 6b), which
might be related to strong observed winds aloft (winds larger than 10 m s$^{-1}$ for these
circumstances, not shown) that transported the sounding farther away from the MWR
encountering different air masses. Despite their coarser resolution, the MWRs were capable of
capturing important gradients in the temperature profile – the existence or lack of surface





temperature inversions at around few hundred meters AGL and the overall decrease in
temperature with height (examples of which are shown in Fig. 6c-d).
To further evaluate how the transition from a stable nighttime to a more convective boundary
layer during the day might affect the accuracy of the temperature observation, the CU MWR
retrieved temperatures were compared to the radiosonde temperatures at different times of the
day (0700 – 1200 LT; 1300 – 1800 LT; 1900 – 2400 LT), as presented in Fig. 7. This figure
contains only the CU MWR because the CU and NOAA MWR were in good agreement over the
two periods presented in Fig. 2, and the CU MWR has a larger dataset because of the outage of
the NOAA MWR in April. No significant differences between these different times of the day
were noticed. For this reason, it can be concluded that the MWR was capable of retrieving
temperatures with a MAE of around 1.2 – 1.3°C during different atmospheric stability
conditions.
In summary below 3.5 km we find consistent behavior of the MWRs among the different months
and similar error statistics for different times of the day using MWR data up to 5 km.

**291    3.2 RASS versus sounding observations**

Temperature observations from the RASSs were compared to radiosonde observations in the
same manner as for the MWRs. As mentioned in section 2.4, we investigate the accuracy of $T_v$
RASS measurements corrected and uncorrected for vertical air motion. Without the vertical
velocity correction (uncorrected $T_v$), no important differences between the three periods of
radiosonde launches emerged (figure not shown). Results of the comparison between uncorrected
$T_v$ measurements from the 915-MHz and the 449-MHz RASS and all the radiosondes launched





in March, April and May are presented in Fig. 8a, b. The MAE for uncorrected $T_v$ observations
was 0.7°C with a bias of 0.2 – 0.3°C (defined as $T_{RASS}$ - $T_{Radiosonde}$).
The impact of the vertical velocity correction is shown in the profile of MAE (Fig. 8c). For
uncorrected $T_v$ (solid lines), MAEs are below 1 °C throughout the entire RASS sampling height.
However, for corrected $T_v$ (dashed lines), MAEs are larger than those for the uncorrected $T_v$, for
both the 915-MHz and 449-MHz RASS, with larger values for the 915-MHz RASS. Similar
results were also found for vertical profiles of the bias (Fig. 8d), for both RASSs. The bias is
around 0.3 °C for the 915-MHz RASS and remains nearly constant with height (solid blue line);
the 449-MHz RASS indicates slightly negative biases below 400 m (solid magenta line),
increasing to around 0.2°C above. For both RASSs, the use of the vertical velocity correction in
the computation of $T_v$ increases the bias substantially (dashed lines), similarly to the impact on
the MAE generated by this correction. This dataset included little convective activity, and so
using the values of $T_v$ corrected for the vertical velocity from RASS measurements is not
beneficial in this study consistent with the results of Görsdorf and Lehmann (2000). Moreover,
the correction is more negative on the 915-MHz RASS $T_v$ which is an indication that the vertical
velocity measurements are more difficult for this system compared to the 449-MHz RASS.

**3.3. MWRs versus *in-situ* tower observations**
In the next step of our assessment, hourly-averaged temperatures from the *in-situ* tower
observations were compared to temperatures derived by the CU MWR for all dates between 9
March – 7 May (Fig. 9). The data set was not divided in different months since the overall
statistics in section 3.1 indicated little variation between the months. The CU MWR is in better
agreement with the tower observations, with a MAE of 0.8 °C (Fig. 9a), than it was with the





sounding observations (MAE= 1.2 °C; Fig. 7a). The MWR temperatures show a positive bias of
0.8 °C compared to the *in-situ* temperature observations. The vertical profile of MAE calculated
between the MWR and *in-situ* temperature observations (Fig. 9b, solid line) indicates higher
values of  ~1 °C at 150 – 250 m, which is exactly the heights where the MAE between the MWR
and the radiosondes showed a local maximum in MAE (Figs. 3c, 4b, 5c). The vertical profile of
bias in temperature between MWR and *in-situ* observations (Fig. 9b, dashed line) show that the
bias is the main contribution to the error, as the value of the bias and of the MAE are very similar
to each other.
While radiosondes were only launched during rain- and snow-free conditions, the comparison
with tower observations (Fig. 9) contains measurements during both times with precipitation and
without precipitation. A comparison between MWR and *in-situ* temperatures observations from
the tower (Fig. 10) shows that the MAE was slightly lower during rainy conditions  (0.8 °C) than
during rain-free conditions (0.9 C), but the overall statistics are not particularly compromised.
Note that we used the rainfall sensor MWRs are equipped with to divide the dataset between
times with and without precipitation.

**3.4. RASS versus *in-situ* tower observations**
Hourly-averaged temperatures from the *in-situ* tower observations were compared to
temperatures derived by the RASSs for all dates between 9 March – 7 May (Fig. 11). Again, the
data set was not divided in different months since the overall statistics in section 3.2 indicated
little variation between the months. Since RASS $T_v$ profiles provided data at different heights
than the tower observations, hourly averaged tower measurements were linearly





interpolated/extrapolated to the 915-MHz RASS's lowest four altitudes (120, 182, 245, and 307
m), and over the 449-MHz RASS's lowest two altitudes (217 and 322 m). As for the comparison
with the radiosondes presented in section 3.2, the effect of applying the correction for the vertical
velocity to the $T_v$ computation by the RASS systems was again investigated.
For the uncorrected $T_v$, the MAE for the RASSs were similar when using the *in situ* tower
observations (Fig. 11a-b) as when using the radiosonde observations (Fig. 8a-b). For bias, both
RASSs slightly underestimated virtual temperatures compared to the tower observations, with a
bias of -0.1 °C for the 915-MHz RASS and -0.4 °C for the 449-MHz RASS. These numbers are
within the expected accuracy of RASS measurements (May et al., 1989).
Vertical profiles of uncorrected $T_v$ MAEs and biases calculated between the tower and both the
915-MHz and 449-MHz RASSs (solid blue and magenta lines in Fig. 11c-d) show more accurate
results than when using the RASS vertical velocity correction (dashed blue and magenta lines).
As previously found in section 3.2, the vertical velocity correction (dashed lines) was not
beneficial to neither the 915-MHz nor the 449-MHz RASS.
We note that comparing Figs. 3-5 to Fig. 8, and Fig. 9 to Fig. 11, the RASS has lower error
statistics than the MWRs which was also shown in Fig. 15 of Lundquist et al. 2016.

**4. Accuracy of the lapse rate**
Several studies have suggested that surface temperature inversions might be smoothed by
remote-sensing instruments with coarse spatial resolutions (Solheim et al. 1998b; Reehorst
2001). Nevertheless, accurate representation of the lapse rate and consequently of atmospheric
stability is essential for wind energy operators to better predict the presence of vertical wind





shear (more likely to happen during stable conditions) and turbulence affecting the load on rotors
(more likely to happen during unstable conditions). Although it is more appropriate to use lapse
rate of potential temperature or virtual potential temperature to provide information on stability
conditions (Friedrich et al, 2012), as a first step we want to compare the ability of MWR with
that of the RASS at evaluating atmospheric stability conditions in the lower boundary layer. To
allow this comparison, we first computed the lapse rate of temperature ($\gamma_T = -dT/dz$) between
50 m and 300 m observed by the CU MWR and compared it with the *in-situ* tower observations
including all dates between 9 March – 7 May (Fig. 12a). Statistics indicate that for the lapse rate
of temperature measured by the CU MWR and the *in-situ* tower measurements the MAE was
about 2.1 ºC km$^{-1}$ with a R$^2$ of 0.91. The same analysis was performed for the lapse rate of virtual
temperature ($\gamma_{T_v} = -dT_v/dz$) computed between the first and fourth level of the 915-MHz RASS
measurements (120-307 m) with the *in-situ* tower observations (Fig. 12b), and for the lapse rate
of virtual temperature ($\gamma_{T_v}$) computed between the first and second level of the 449-MHz RASS
measurements (217-322 m) with the *in-situ* tower observations (Fig. 12c). To have a compatible
comparison between the ability of the MWR at measuring lapse rate with that of the RASSs we
computed the same statistics (MAE, bias, R$^2$, slope) presented in Fig. 12a, but first interpolating
the CU MWR observations over the heights covered by the 915-MHz RASS over the tower
measurements (120-307 m), and later interpolating the MWR observations over the heights
covered by the 449-MHz RASS over the tower measurements (217-322 m). The first gave a R$^2$ =
0.89 for the CU MWR and R$^2$ = 0.81 for the 915-MHz RASS, while the second gave a R$^2$ = 0.79
for the CU MWR and R$^2$ = 0.6 for the 449-MHz RASS, resulting in the best R$^2$ for the MWR.
In addition to the lapse rates of temperature ($\gamma_T$), we calculated lapse rate of potential
temperature from CU MWR measurements, as $\gamma_\theta = -d\theta/dz$ (differences with the lapse rate of





virtual potential temperature were practically unnoticeably). The statistics (MAE, bias, $R^2$, slope)
were calculated for $\gamma_\theta$ using different tower levels and the results are presented in Table 2. We
note that the agreement between the lapse rate of potential temperature measured by the CU
MWR and the *in-situ* tower measurements is best when it is computed between 50 m and 300 m
(larger $dz$), with a coefficient of determination of 0.91. A comparison between the time series of
$\gamma_\theta$ (between 50 and 300 m) as computed by the *in-situ* tower measurements and as computed by
the CU MWR is presented in Fig. 13 for all dates between 9 March – 7 May. The CU MWR
follows the diurnal cycle of $d\theta/dz$ quite well, with the largest differences occurring at the
minimum and maximum values.
To better quantify the differences in temperature between the CU MWR and the *in-situ*
observations, the data set was finally divided into times when the atmosphere was stable
($d\theta/dz \geq 0$) and unstable ($d\theta/dz < 0$), based on the observations conducted by the CU MWR
presented in Fig. 13. Temperatures observed by the CU MWR were compared during stable and
unstable conditions to *in-situ* tower observations at six height levels between 50 – 300 m.
Smaller MAE's occurred in unstable conditions (MAE = 0.8 $^{\circ}$C; Fig. 14a) compared to stable
conditions (MAE = 1.2 $^{\circ}$C; Fig. 14b). Similarly the bias was smaller in unstable conditions
compared to stable, and $R^2$ was larger.

**5. Accuracy of the MWR water vapor density and humidity profiles**
Differences in water vapor density (*WVD*) and relative humidity (*RH*) between the two MWRs
were analyzed before comparing to sounding observations. Profiles derived from averaged 15$^{\circ}$
elevation scans between the surface and 10 m AGL were compared during the time periods when
both instruments were functioning (9/3/2015 – 4/4/2015 in Fig. 15a-c; 28/4/2015 – 7/5/2015 in





Fig. 15b-d). For the *WVD* comparison, the MAE's for the two systems were 0.1 and 0.2 g m$^{-3}$,
the biases were ~ 0.1 g m$^{-3}$ and R$^2$ was very close to 1 (Fig. 15a-b). For the *RH* comparison, the
MAE's were 4.1 and 4.8%, the biases were 2.1 and 0.9%, while the coefficients of determination
were both 0.96 (Fig. 15c-d). The values of bias, R$^2$, and slope indicated a good agreement
between the instruments over the periods during which they were both functioning properly.
Lastly, water vapor density and relative humidity derived from the MWRs between the surface
and 5 km AGL are compared to radiosonde observations from March, April, and May (Figs. 16-
19 for *WVD*; Figs. 20-23 for *RH*). For *WVD*, in March and April, MAE for both instruments
show values equal to 0.3 g m$^{-3}$, also reported by  Cimini et al. 2011, the bias was close to 0 g m$^{-3}$
and the coefficient of determination was 0.92 (Fig. 16a-b and Fig. 17a). Vertical profiles of MAE
(Fig. 16c and Fig. 17b) show values of about 0.3 – 0.2 g m$^{-3}$ up to 3 – 3.5 km, with decreasing
MAE above 3.5 km. Larger MAEs were observed for *WVD* in May (Fig. 18) compared to March
and April. MAE is equal to 0.5 g m$^{-3}$ (Fig. 18a-b) with R$^2$ values of about 0.92. Vertical profiles
of MAE in May indicate larger values below 2.5 km, where *WVD* profiles from the sounding
showed more variability, decreasing above. Overall MWRs are able to follow the radiosonde
vertical profile of *WVD* as presented in Fig. 19, although some information is missed due to the
coarser MWR resolution compared to the sounding observations.
For *RH*, MAE for both instruments show values below 10 % in March (Fig. 20a-b). A relatively
large scatter (R$^2$ of ~0.8) is an indication of large variation in relative humidity. Some of the
variability and associated large scatter might be attributed to the sounding encountering different
air masses or even clouds at higher altitudes, as indicated by the vertical profiles of MAE. These
profiles show that the MAE's are about 5 – 8% below ~1 km, with the MAE continuously
increasing with increasing height (Fig. 20c). Larger MAEs were observed in April and May
(Figs. 21-22) compared to March (Fig. 20). MAE's range between 11 – 14% with $R^2$ values of
about 0.5 (Fig. 21a and Fig. 22a-b). Vertical profiles of MAE in April and May indicate a similar
pattern compared to March. Lower values (5 – 12%) were observed below ~1 km, while larger
values occurred around 1 km and between 3 – 4 km. Since the three-dimensional humidity field
is highly variable and strongly depends whether or not the instruments (both MWR and
sounding) encountered clouds, the large MAE's between 1 – 4 km are most likely due to changes
in air mass or the existence of clouds.
High-resolution soundings with vertical resolution of few meters show much more detail
compared to the smooth MWR humidity profiles, as seen in Fig. 23. These examples show that
while the MWRs are capable of reproducing the general trend compared to sounding
observations, differences between the MWR and the sounding can be as high as 20-25%.

**6. Conclusions**
Data collected during the XPIA campaign in spring 2015 were used to assess the accuracy of
temperature, water vapor density, and relative humidity profiles from two MWRs, one 915-MHz
WPR-RASS system, and one 449-MHz WPR-RASS system with respect to *in-situ* reference
measurements from 61 radiosonde launches and temperature and relative humidity
measurements at six different levels from a 300-m co-located tower. Results indicate a mean
absolute error in the temperature retrieved by the MWRs below 1.5 °C for the layer of the
atmosphere up to 5 km.  However, the details of the inversions were consistently in error, with
the MWRs too cold at the surface and too warm above 250 m. Our results revealed that the
overall statistics for MWRs temperature measurements were slightly better for unstable
conditions than stable, while the overall statistics for MWR temperature measurements were not





particularly compromised during rainy conditions, compared to rain-free conditions. In addition
we find consistent behavior of the MWRs among the different months and similar error statistics
for different times of the day.
For the RASSs we found a mean absolute error in the virtual temperature below 0.8 °C in the
layer of the atmosphere covered by these measurements (up to approximately 1.6 – 2 km) and
that using the values of $T_v$ corrected for the vertical velocity can decrease temperature accuracy,
and should only be used with caution. For this dataset, the correction for the vertical velocity
applied to calculate $T_v$ was not beneficial to the accuracy of RASS measurements of $T_v$ under any
weather condition. In general the RASSs have overall lower error statistics than the MWRs for
the layer of the atmosphere covered by the RASSs.
We additionally assessed the accuracy of these remote-sensing instruments at measuring
atmospheric stability conditions in the lower boundary layer, finding a coefficient of
determination between the lapse rate measured by the MWR and the tower measurements over
the tower levels between 50 and 300 m ranged from 0.76 to 0.91, with the best value (0.91)
found when the lapse rate is computed between 50 m and 300 m (larger $dz$).  These positive
results demonstrate that profiling microwave radiometers can be useful for understanding
conditions that can lead to strong vertical wind shear or turbulence, which can affect the load on
rotors.
We also assessed the accuracy of MWRs at retrieving water vapor density profiles, finding a
mean absolute error below 0.5 g m$^{-3}$ for the layer of the atmosphere up to 5 km.
Finally, our study unsurprisingly revealed that relative humidity profiles measured by the MWR
lack high resolution details compared to radiosonde measurements with differences between the
MWR and the sounding that can be as high as 20-25% and in average, for the layer of the



atmosphere up to 5 km, of the order of 8-14%. For this reason, our future research will utilize the
unique dataset collected for XPIA to combine the information obtained from WPR potential
refractivity profiles and from MWR potential temperature profiles to improve the accuracy of
atmospheric humidity profiles (Bianco et al., 2005).


**Acknowledgements**
The authors wish to acknowledge Alex St. Pé from University of Maryland, Baltimore County,
as well as the University of Colorado Boulder students Joseph C.-Y. Lee, Paul Quelet, Clara St.
Martin, Brian Vanderwende, Rochelle Worsnop who launched the radiosondes and Mr. Timothy
Lim of NCAR for providing them careful training. We are very grateful for the assistance
received from Radiometrics Corporation in setting up and comparing the two MWRs. Thank
goes to the University of Colorado Boulder students Joshua Aikins and Evan Kalina who set up
the CU radiometer. We express appreciation to the National Science Foundation for supporting
the CABL deployments (https://www.eol.ucar.edu/field_projects/cabl) of the MGAUS
radiosondes and the tower instrumentation. We thank Lefthand Water District for providing
access to the Water Tank site. We would like to acknowledge operational, technical, and
scientific support provided by NCAR's Earth Observing Laboratory, sponsored by the National
Science Foundation, as well as by Timothy Coleman of NOAA/PSD. Partial support for the
UMBC deployments was provided by the Maryland Energy Administration. NREL is a national
laboratory of the U. S. Department of Energy, Office of Energy Efficiency and Renewable
Energy, operated by the Alliance for Sustainable Energy, LLC. Funding for this work was
provided by the U. S. Department of Energy, Office of Energy Efficiency and Renewable



Energy, Wind and Water Power Technologies Office, and by NOAA's Earth System Research
Laboratory.























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



**Table 1. Statistical values for the NOAA and CU MWRs vs radiosonde observations of $T$**
**for the three periods of radiosonde launches and for the zenith and at 15º off-zenith angles.**
**Bias and MAE are in (ºC).**

| | March (38 radiosonde) | | | | April (13 radiosonde) | | | | May (10 radiosonde) | | | |
|---|---|---|---|---|---|---|---|---|---|---|---|---|
| | Bias | MAE | $R^2$ | slope | Bias | MAE | $R^2$ | slope | Bias | MAE | $R^2$ | slope |
| CU MWR (zenith) vs radiosonde | -0.3 | 1.5 | 0.98 | 0.91 | 0.2 | 1.1 | 0.99 | 0.94 | 0.2 | 1.1 | 0.99 | 0.93 |
| CU MWR (15º elevation) vs radiosonde | -0.8 | 1.5 | 0.98 | 0.93 | -0.2 | 0.9 | 0.99 | 0.97 | -0.3 | 1.1 | 0.98 | 0.97 |
| NOAA MWR (zenith) vs radiosonde | -0.5 | 1.4 | 0.98 | 1.03 | Missing data | | | | -0.1 | 1 | 0.99 | 1.04 |
| NOAA MWR (15º elevation) vs radiosonde | 0 | 1.3 | 0.98 | 0.94 | | | | | 0.4 | 1 | 0.98 | 0.97 |











**Table 2. Statistical values for the CU MWRs vs tower observations of lapse rate ($\gamma_\theta =$**
**$-d\theta/dz$) for different tower levels. Bias and MAE are in (ºC km$^{-1}$).**

| Lapse rate ($\gamma_\theta = -d\theta/dz$) Between 50 - 150 m | | | | Lapse rate ($\gamma_\theta = -d\theta/dz$) Between 50 - 200 m | | | | Lapse rate ($\gamma_\theta = -d\theta/dz$) Between 50 - 250 m | | | | Lapse rate ($\gamma_\theta = -d\theta/dz$) Between 50 - 300 m | | | |
|---|---|---|---|---|---|---|---|---|---|---|---|---|---|---|---|
| Bias | MAE | $R^2$ | slope | Bias | MAE | $R^2$ | slope | Bias | MAE | $R^2$ | slope | Bias | MAE | $R^2$ | slope |
| 0.76 | 5.7 | 0.76 | 0.59 | 0.52 | 4.0 | 0.82 | 0.70 | -0.19 | 3.0 | 0.88 | 0.81 | -0.1 | 2.2 | 0.91 | 0.96 |
















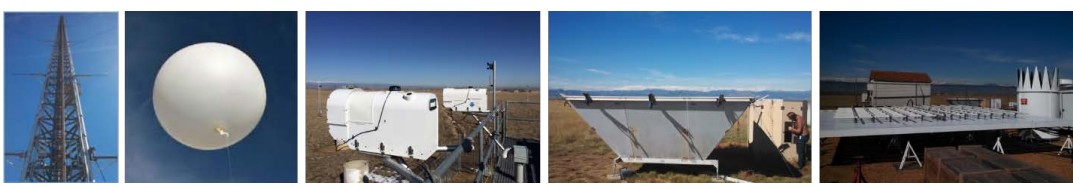

**Figure 1:  Instruments used in this study. From the left: 300-m equipped meteorological**
**tower (photo credit: Katie McCaffrey), radiosonde, 2 MWRs, 915-MHz and RASS system**
**(photo credit: Katie McCaffrey), 449-MHz and RASS system (photo credit: Katie**
**McCaffrey).**





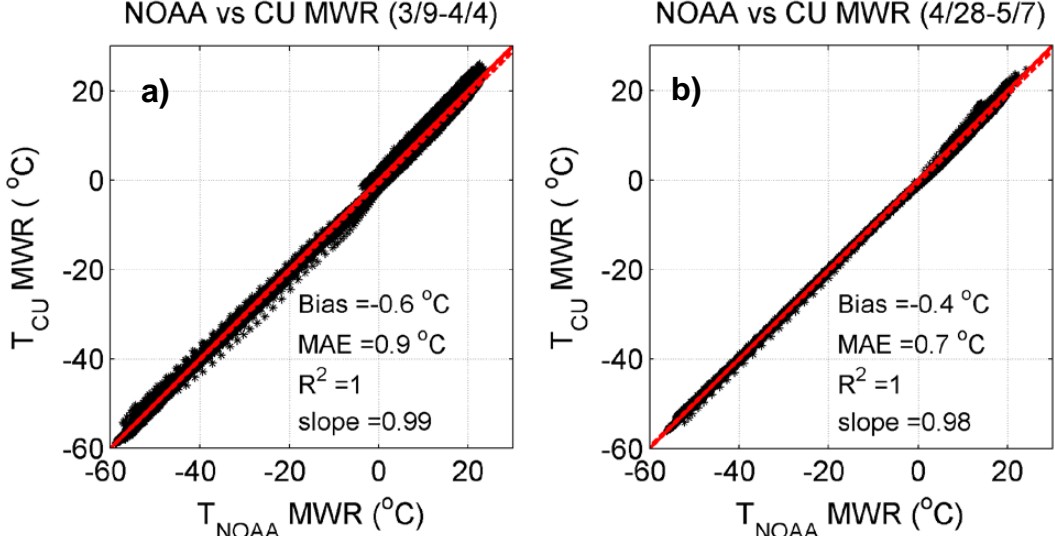

**Figure 2:** **Comparison between temperature observed by CU MWR and NOAA MWR**

**between: a) 9 March – 4 April, and b) 28 April – 7 May, 2015. The missing days in April**

**coincide with the failure of the NOAA MWR surface sensor. A 1-to-1 line is indicated in**

**solid red, and the regression is shown by the dashed red line.**





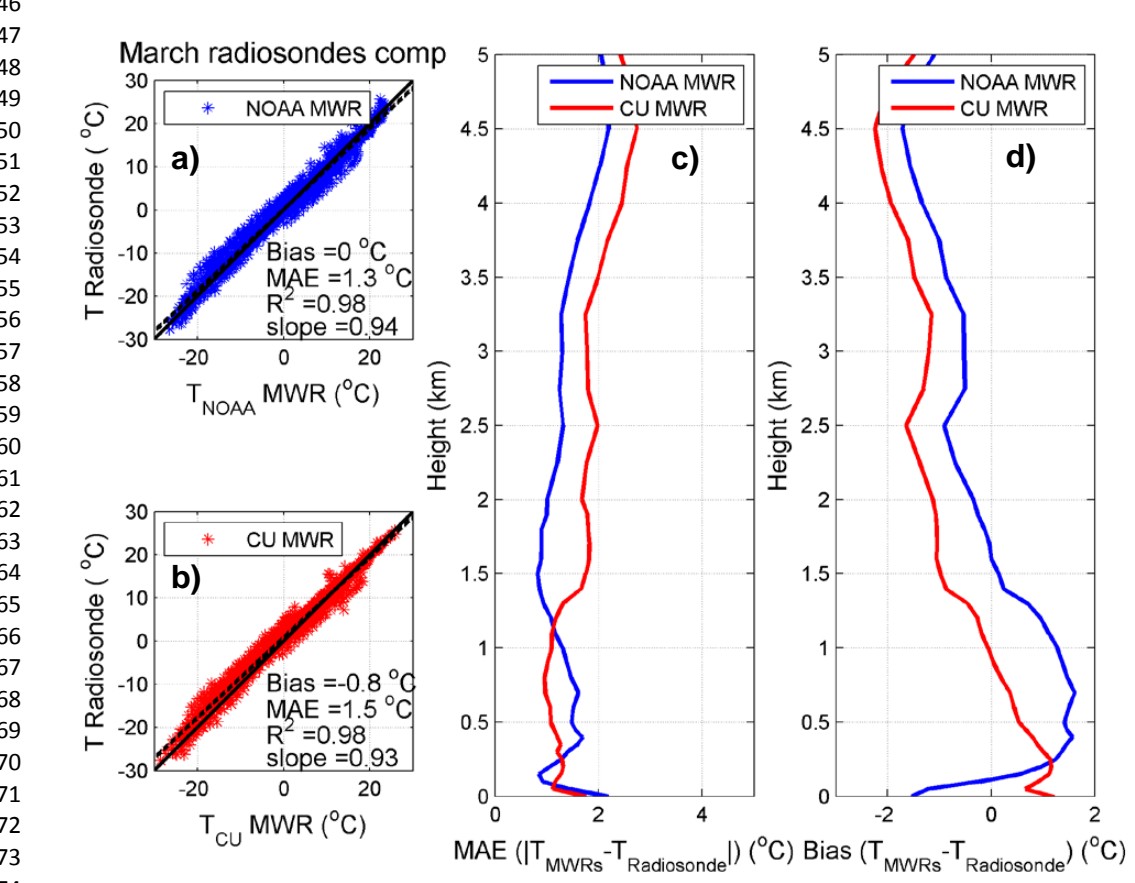

**Figure 3: MWRs vs radiosonde comparison of *T* for 9 – 19 March including 38 radiosonde launches: a)-b) One-to-one comparisons between *T* observed by the radiosondes and the a) NOAA and b) CU MWR between the surface and 5 km AGL. One-on-one line is indicated as solid black line and the regression as dashed black line. c)-d) Vertical profiles of MAE and Bias for the same variable for the NOAA MWR (blue line) and CU MWR (red line).**














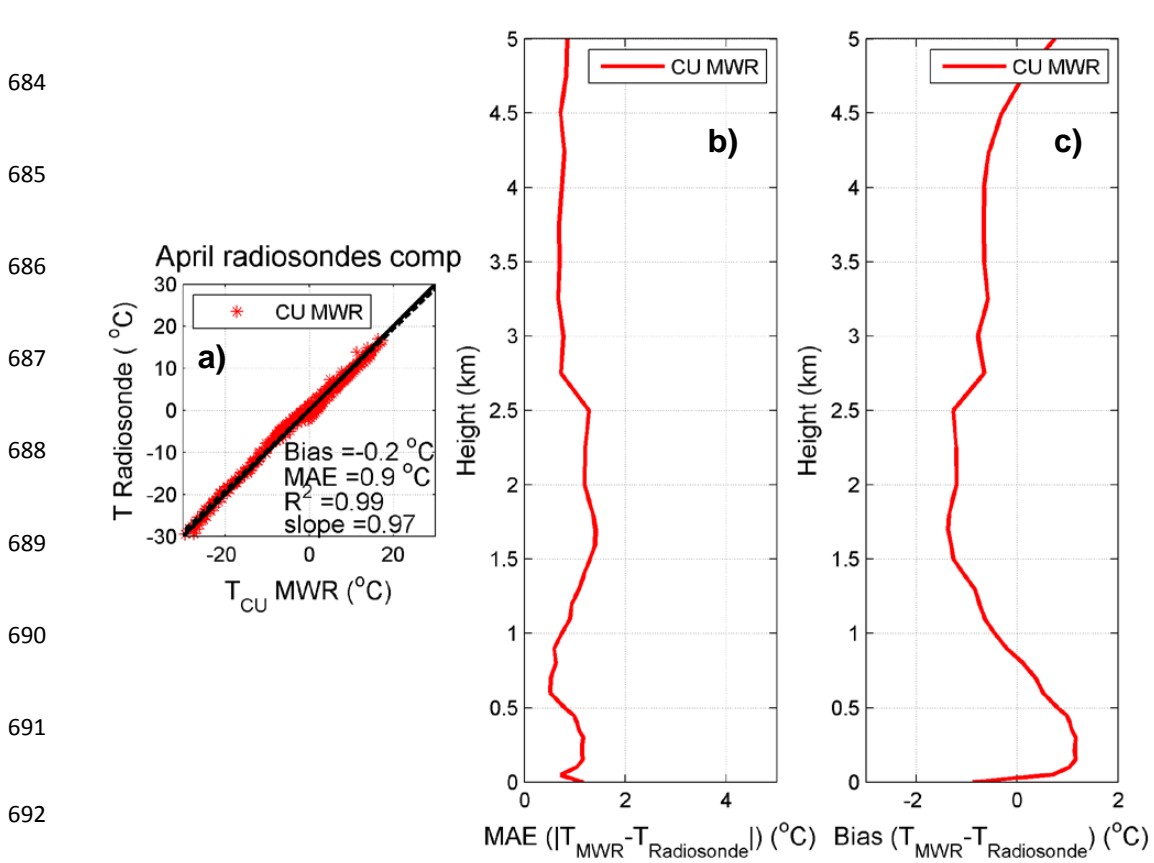

**Figure 4: Same as in Fig. 3, but for 15 and 20 – 22 April including 10 radiosonde launches.**


**Note that the pressure sensor of the NOAA MWR was broken between 5 – 27 April,**


**therefore the NOAA MWR vs radiosonde comparison (*T*) over this period is not presented.**




















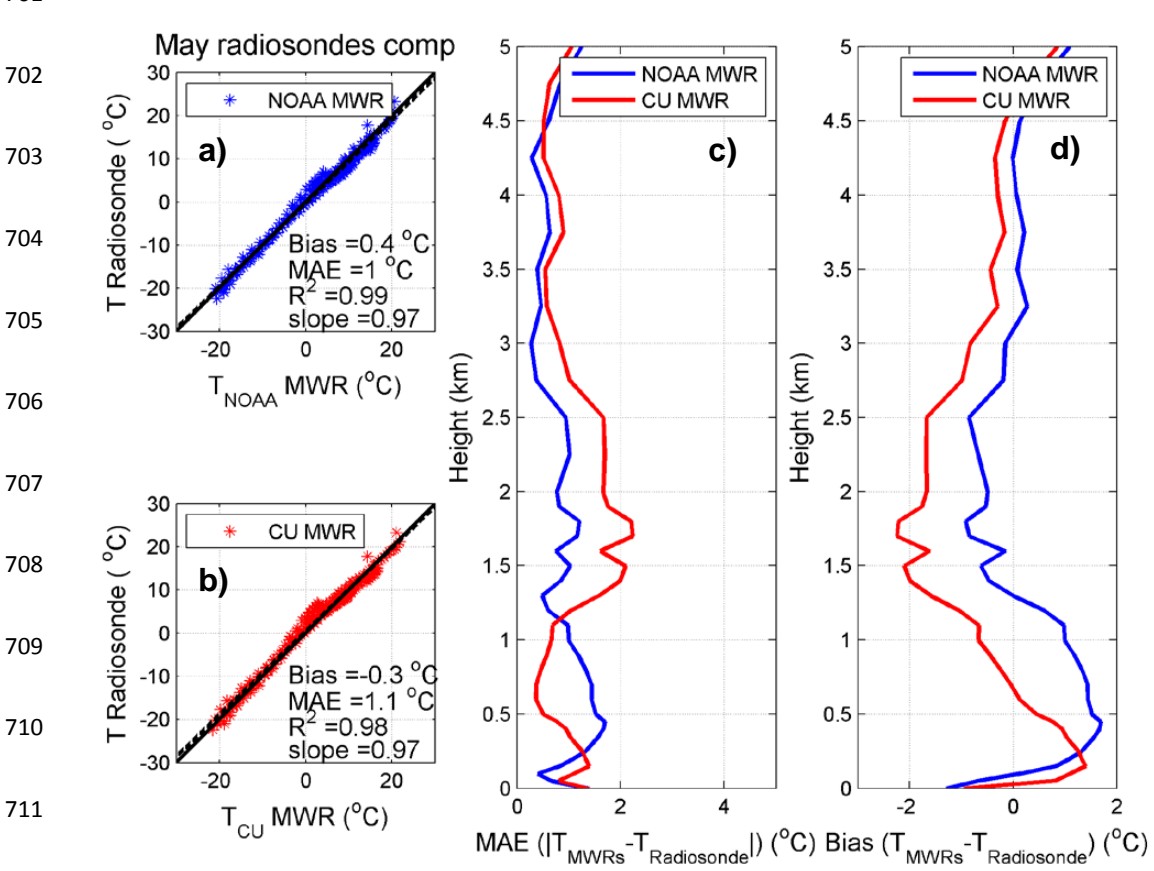

**Figure 5:  Same as in Fig. 3, but for the May period of 13 radiosonde launches.**










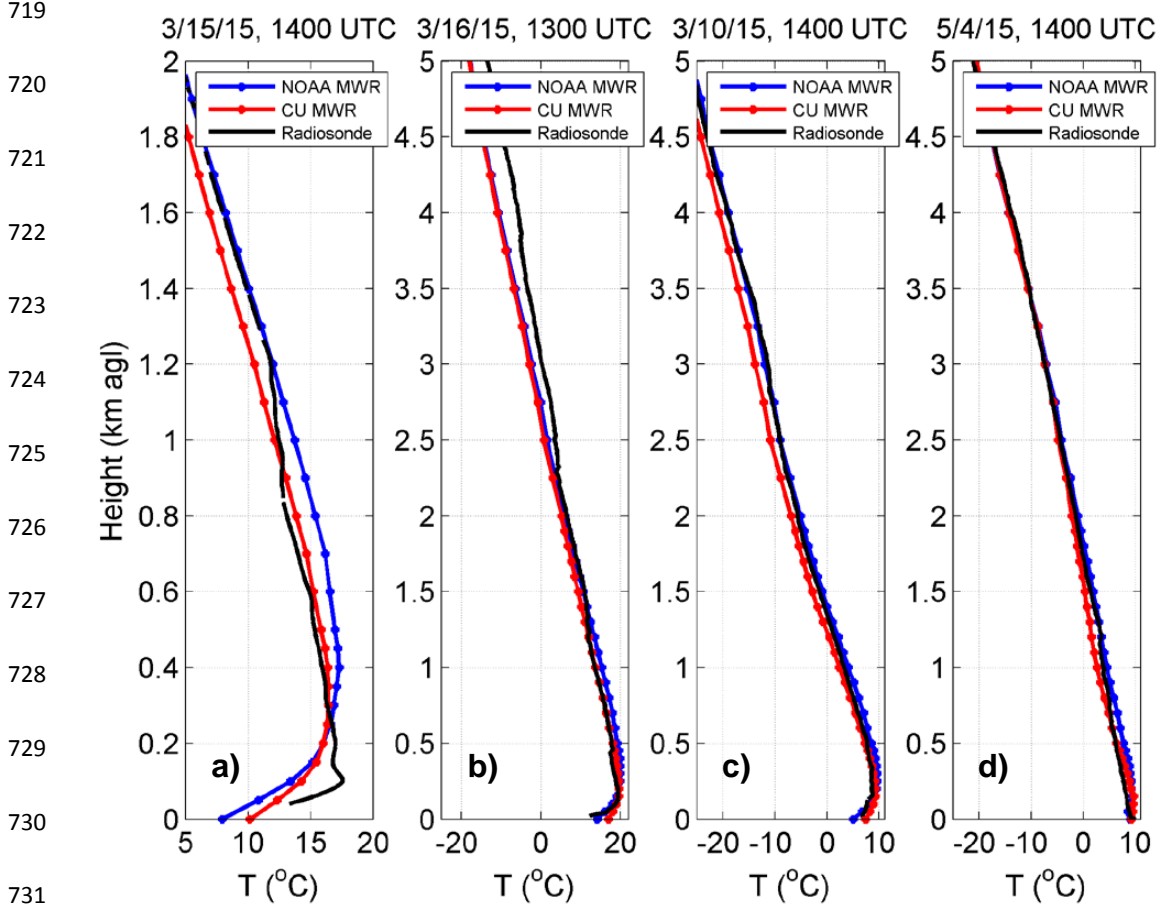


**Figure 6: Vertical profiles of temperature as observed by MWRs (blue line: NOAA MWR;**
**red line: CU MWR) and radiosonde (black line) at: a) 0800 LT (1400 UTC) on 15 March,**
**b) 0700 LT (1300 UTC) on 16 March, and c) 0800 LT (1400 UTC) on 10 March 2015, d)**
**0800 LT (1400 UTC) on 4 May 2015.**




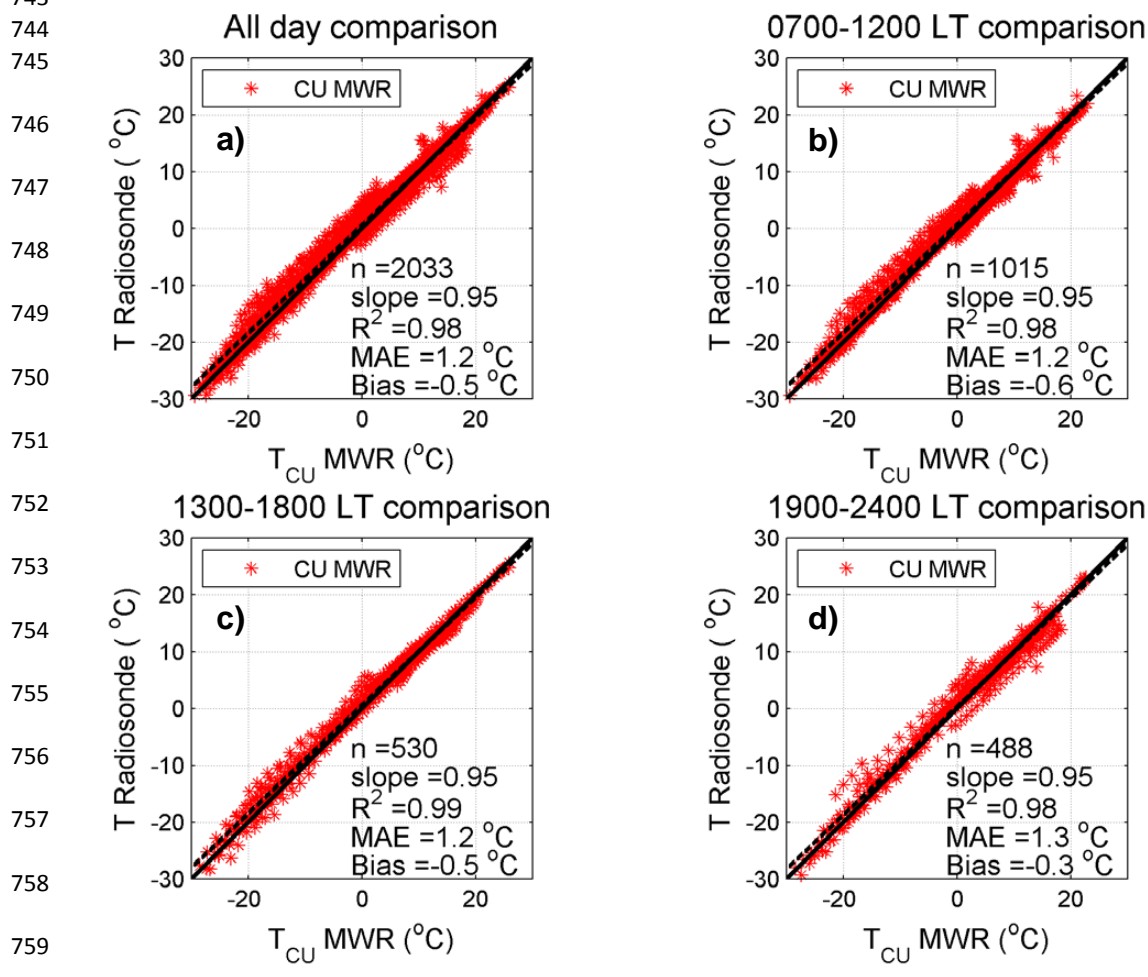

**Figure 7:  CU MWR vs radiosonde comparison of temperature over: a) 0700 – 2400 LT, b) 0700 – 1200 LT, c) 1300 – 1800 LT, and d) 1900 – 2400 LT between the surface and 5 km AGL. Data were collected on 9 – 19 March (38 soundings), 15 April and 20 – 22 April (10 soundings), and 1 – 4 May (13 soundings). Note that no radiosonde were launched between 0100-0600 LT. One-on-one line is indicated as solid black line and the regression as dashed black line.**





767

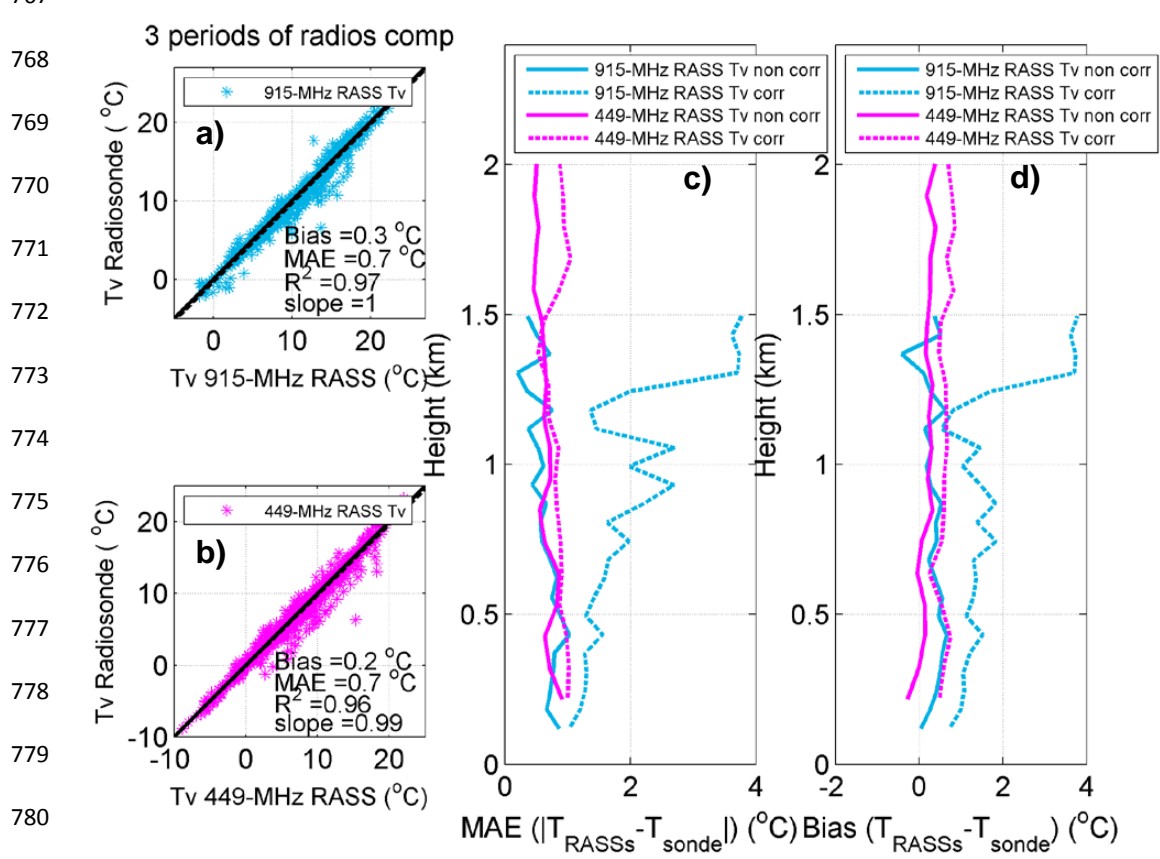

781

**Figure 8: 915-MHz RASS (light blue) and 449-MHz RASS (magenta) vs radiosonde**

**comparison of *Tv* over the 3 periods (March, April, and May) of radiosonde launches**

**combined together. a)-b) One-to-one comparison between radiosonde and a) 915-MHz**

**between 120 m and ~1.6 km AGL and b) 449-MHz RASS between 217 m – ~2 km AGL.**

**The correction for the vertical velocity was NOT applied. One-on-one line is indicated as**

**solid black line and the regression as dashed black line. c)-d) Vertical profiles of MAE and**

**Bias for *Tv* with (dashed lines) and without (solid lines) vertical velocity correction.**





















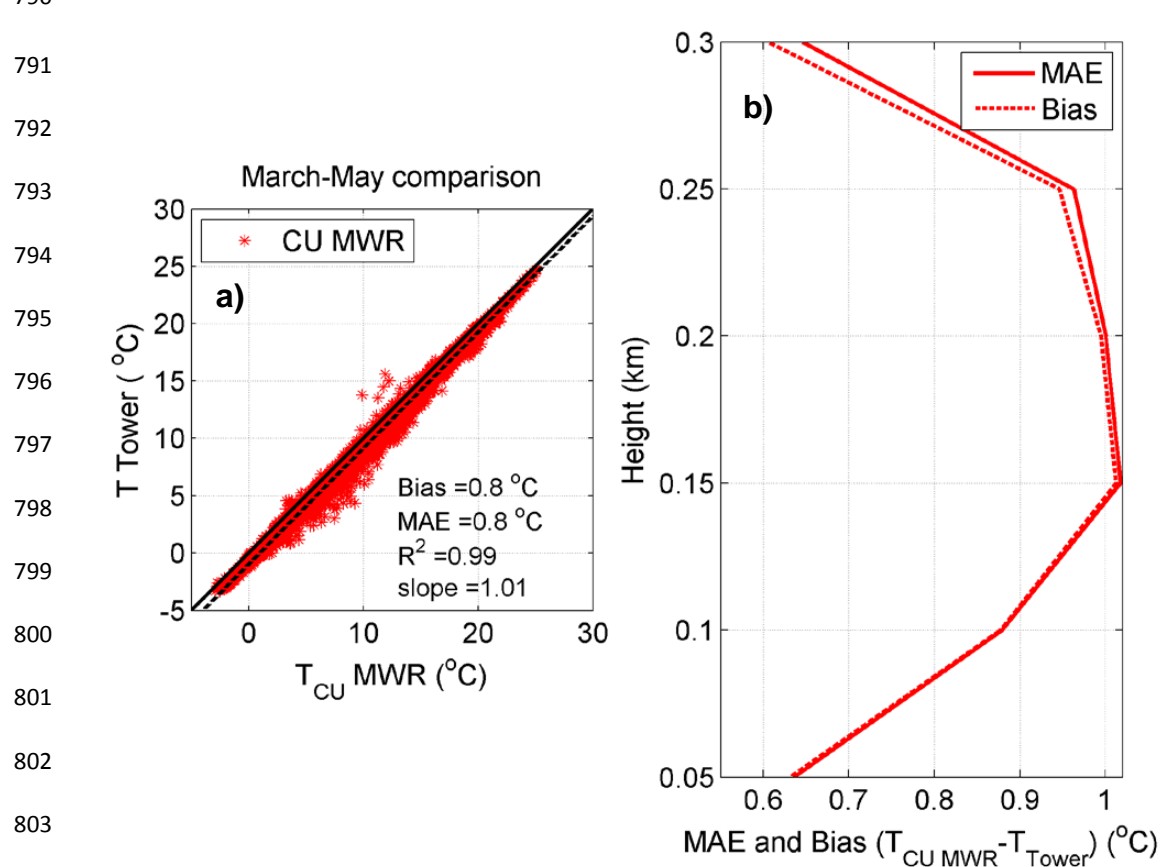

**Figure 9: CU MWR vs tower comparison of temperature for all dates between 9 March – 7**
**May. a) One-to-one comparison. One-on-one line is indicated as solid black solid line and**
**the regression as dashed black line. b) Vertical profiles of MAE and Bias for the same**
**variable. Temperatures were observed at the tower at 50, 100, 150, 200, 250, and 300 m**
**AGL, which collocates with MWR levels.**



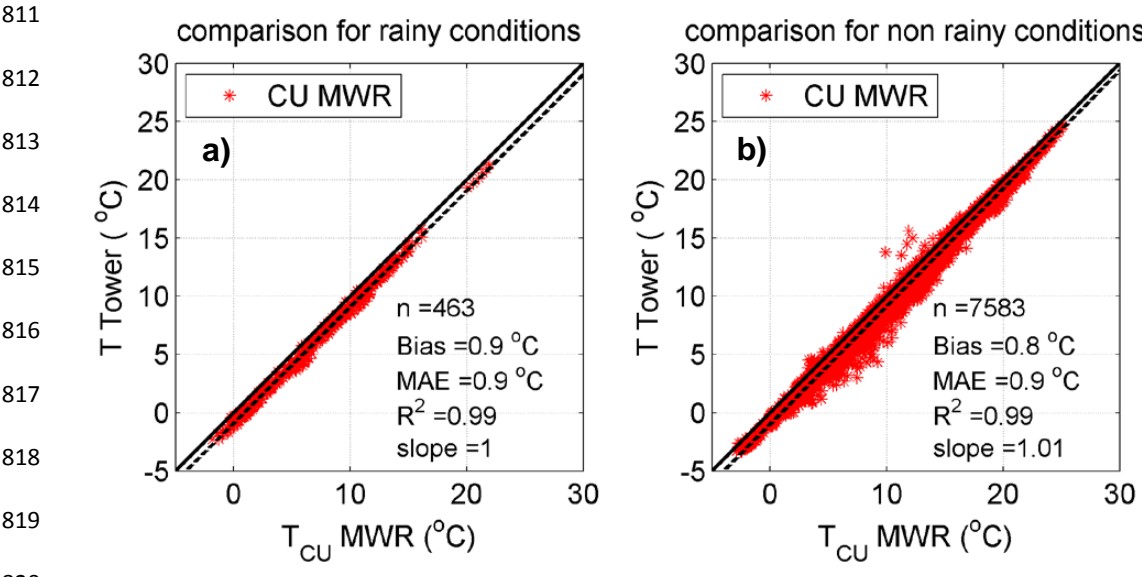

**Figure 10: CU MWR vs tower temperature measurements during: a) rainy conditions, b) no-rain conditions as measured by the CU MWR. One-on-one line is indicated as black solid line and the regression as dashed black line.**

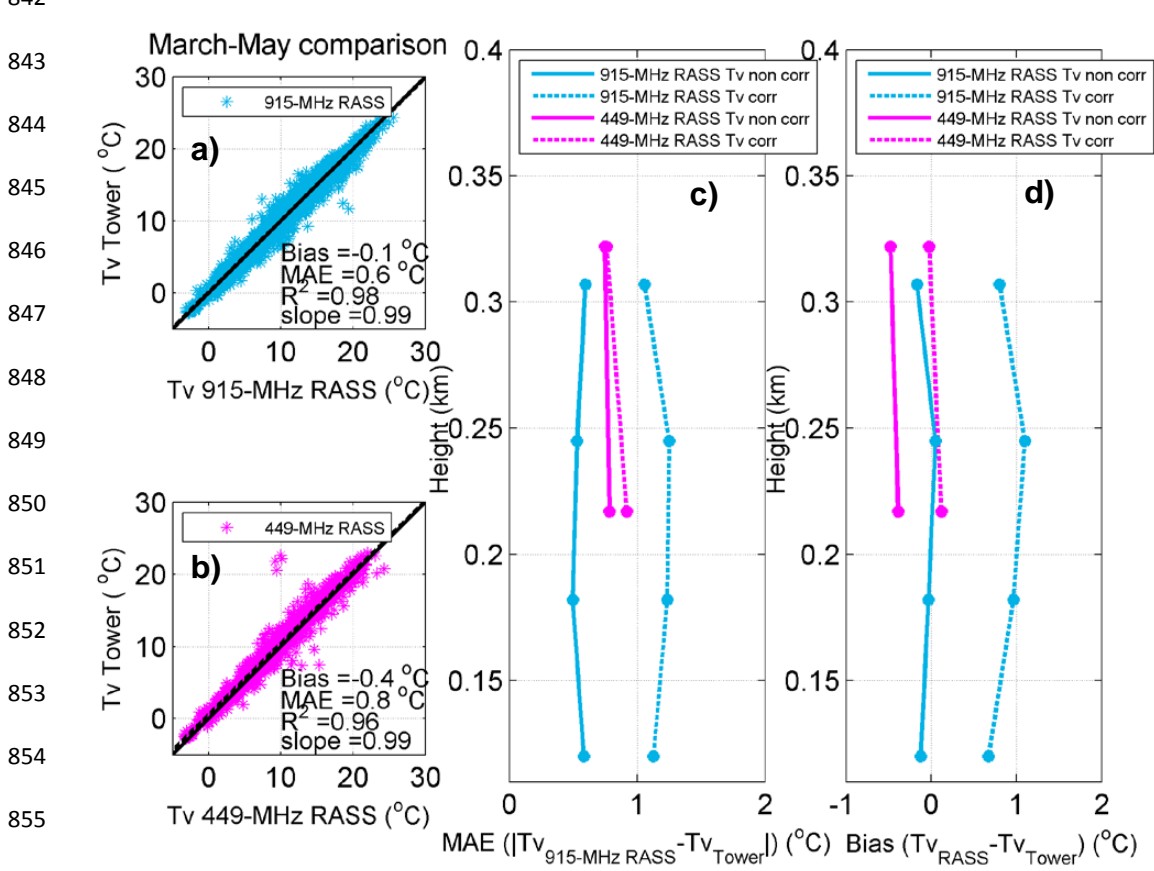

**Figure 11: 915-MHz (light blue) and 449-MHz (magenta) RASS vs tower comparison of $T_v$ for all dates between 9 March – 7 May. a-b) One-to-one comparisons between in-situ tower observations and uncorrected $T_v$ observations. One-on-one line is indicated as solid black line and the regression as dashed black line. c)-d) Vertical profiles of MAE and Bias for $T_v$ with (dashed lines) and without (solid lines) the vertical velocity correction. Height is AGL.**





86
86
86
86
86
87
87
87
87
87
87
87
87
87
87
87,

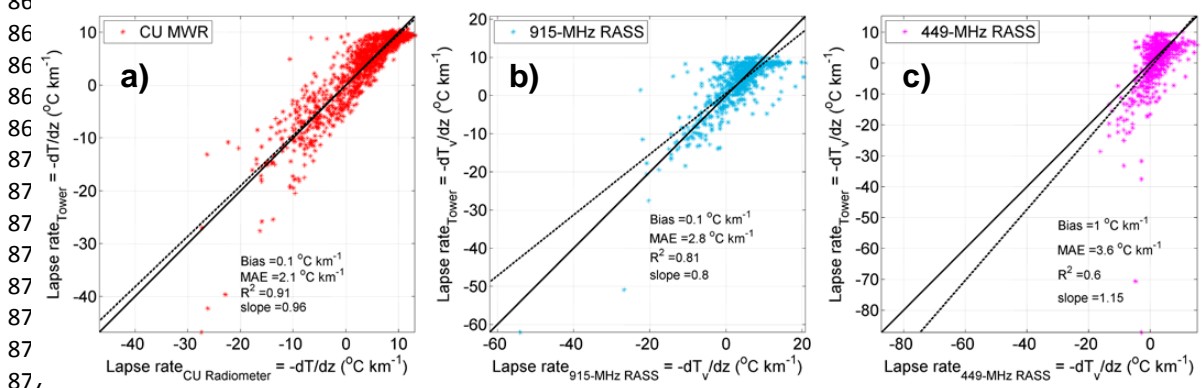

**Figure 12: Comparison of atmospheric lapse rate for all dates between 9 March – 7 May,**
**2015 for: a) CU MWR vs tower (between first and last level of the tower measurements, 50-**
**300m), b) 915-MHz RASS vs tower (between first and fourth level of the 915-MHz RASS**
**measurements, 120-307m), c) 449-MHz RASS vs tower (between first and second level of**
**the 915-MHz RASS measurements, 217-322m). Negative lapse rate represents stable**
**atmospheric conditions. One-on-one line is indicated as solid black line and the regression**
**as dashed black line.**





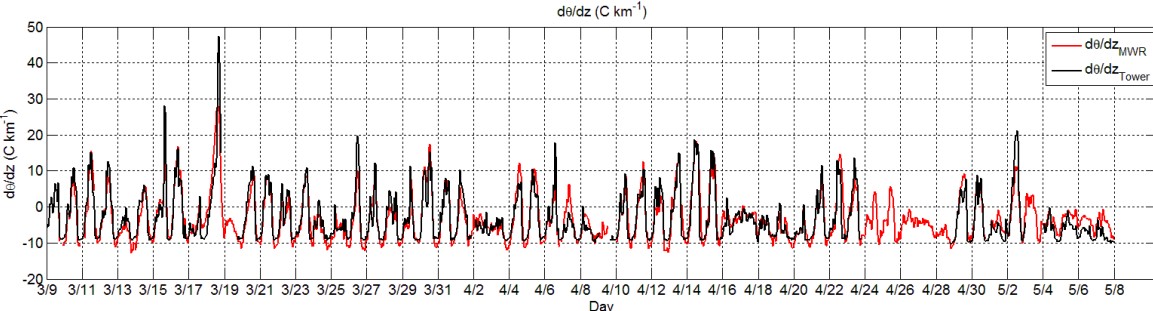


**Figure 13: Lapse rate of potential temperature between 50 and 300 m AGL derived from**

**observations conducted by the CU MWR (red line) and from tower observations (black**

**line) for all dates between 9 March – 7 May.**




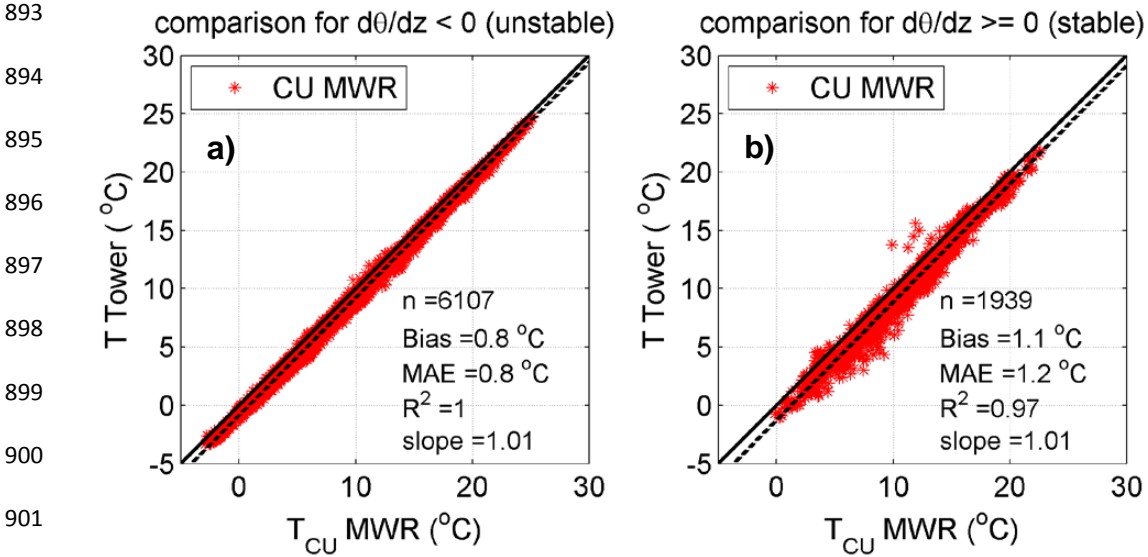

**Figure 14: CU MWR vs tower comparison of T for all dates between 9 March – 7 May, for: a)** $d\vartheta/dz < 0$**, b)** $d\vartheta/dz \geq 0$ **between 50 – 300 m. Stability was determined by temperature differences measured by the CU MWR. One-on-one line is indicated as solid black line and the regression as dashed black line.**





**Figure 15: Comparison between a-b)** *WVD* **and c-d)** *RH* **observed by CU MWR and NOAA**

**MWR between 9 March – 4 April 2015 (a and c) and 28 April – 7 May 2015 (b and d). The**

**missing days in April coincide with the failure of the NOAA MWR surface sensor. One-on-**

**one line is indicated as solid red line and the regression as dashed red line.**





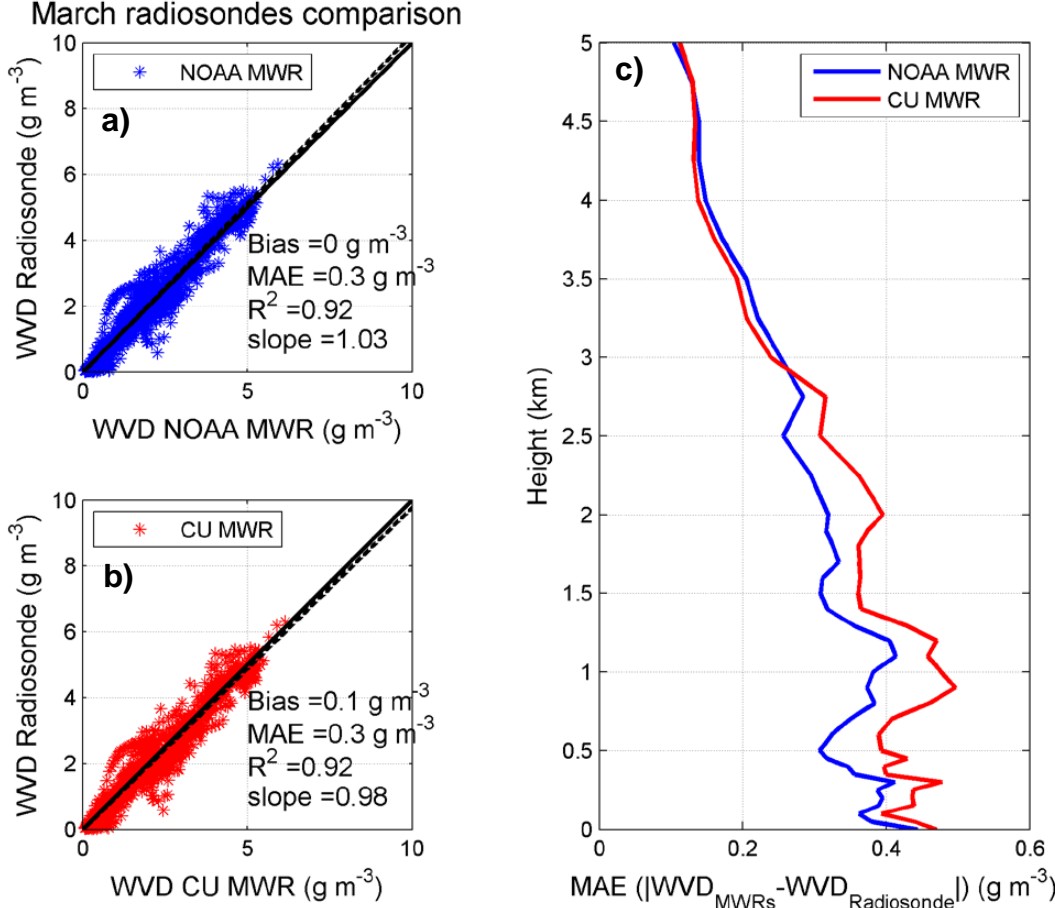



**Figure 16: MWR vs radiosonde comparison of *WVD* over the March period of 38**
**radiosonde launches. a)-b) are one-to-one comparisons of *WVD* observed by the**
**radiosondes and the a) NOAA and b) CU MWR between the surface and 5 km AGL. One-**
**on-one line is indicated as solid black line and the regression as dashed black line. c)**
**Vertical profiles of MAE for the same variable.**




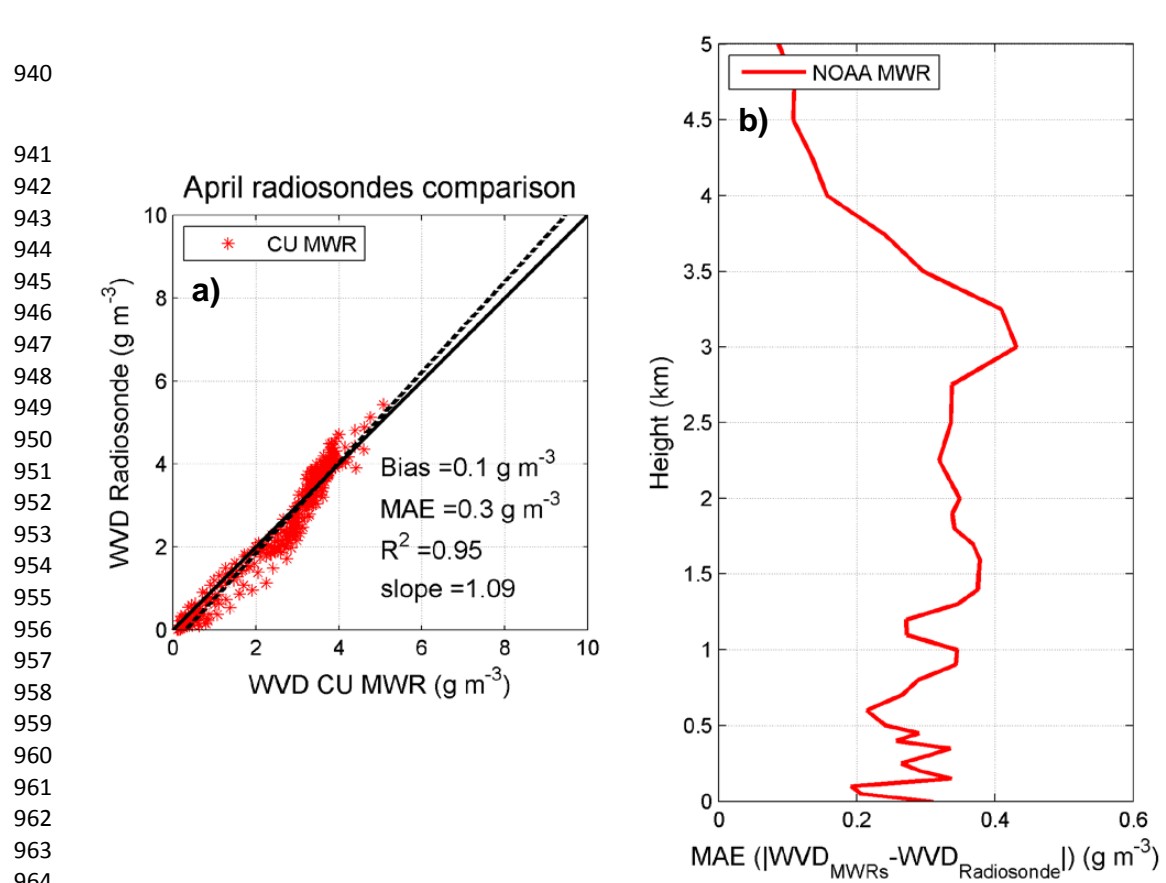

**Figure 17:  Same as in Fig. 16, but for 15 and 20 – 22 April including 10 radiosonde**
**launches. Note that the pressure sensor of the NOAA MWR was broken between 5 – 27**
**April, therefore the NOAA MWR vs radiosonde comparison (*WVD*) over this period is not**
**presented.**



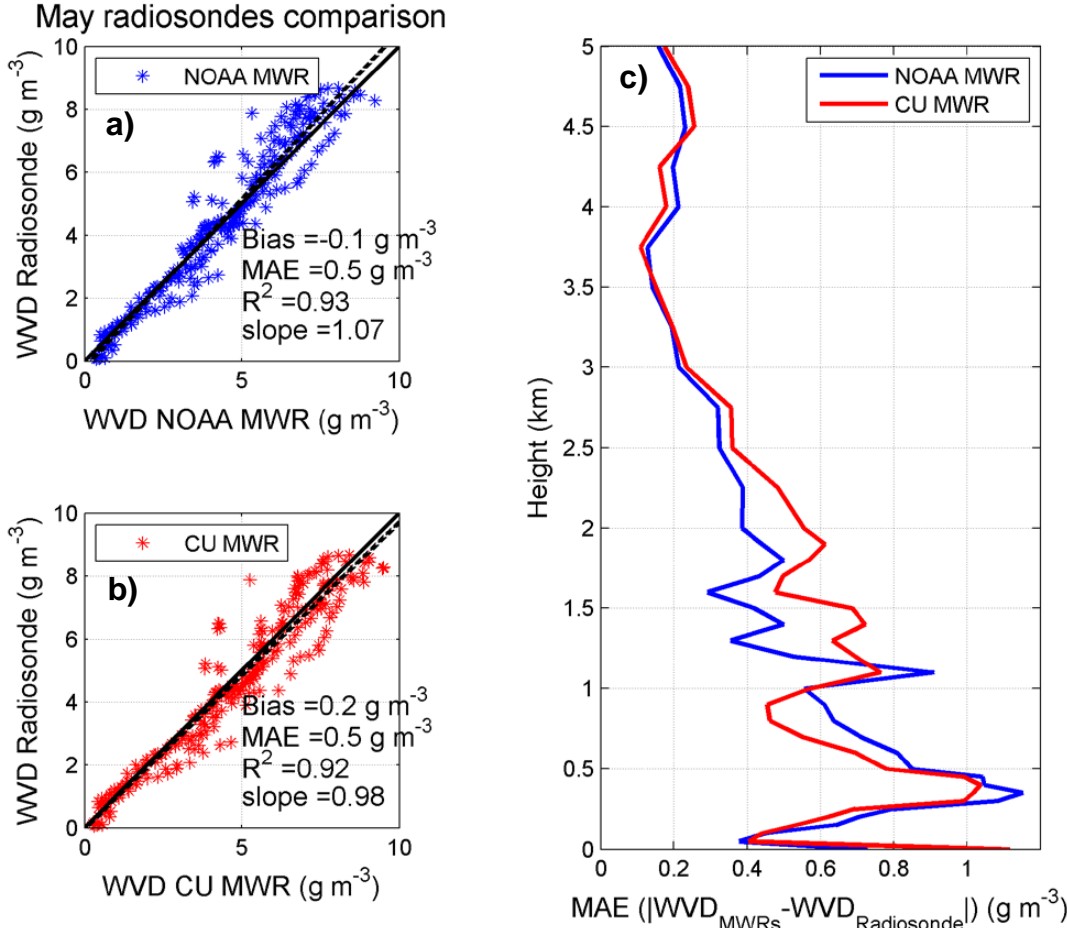

**Figure 18:  Same as in Fig. 16, but for the May period of 13 radiosonde launches.**




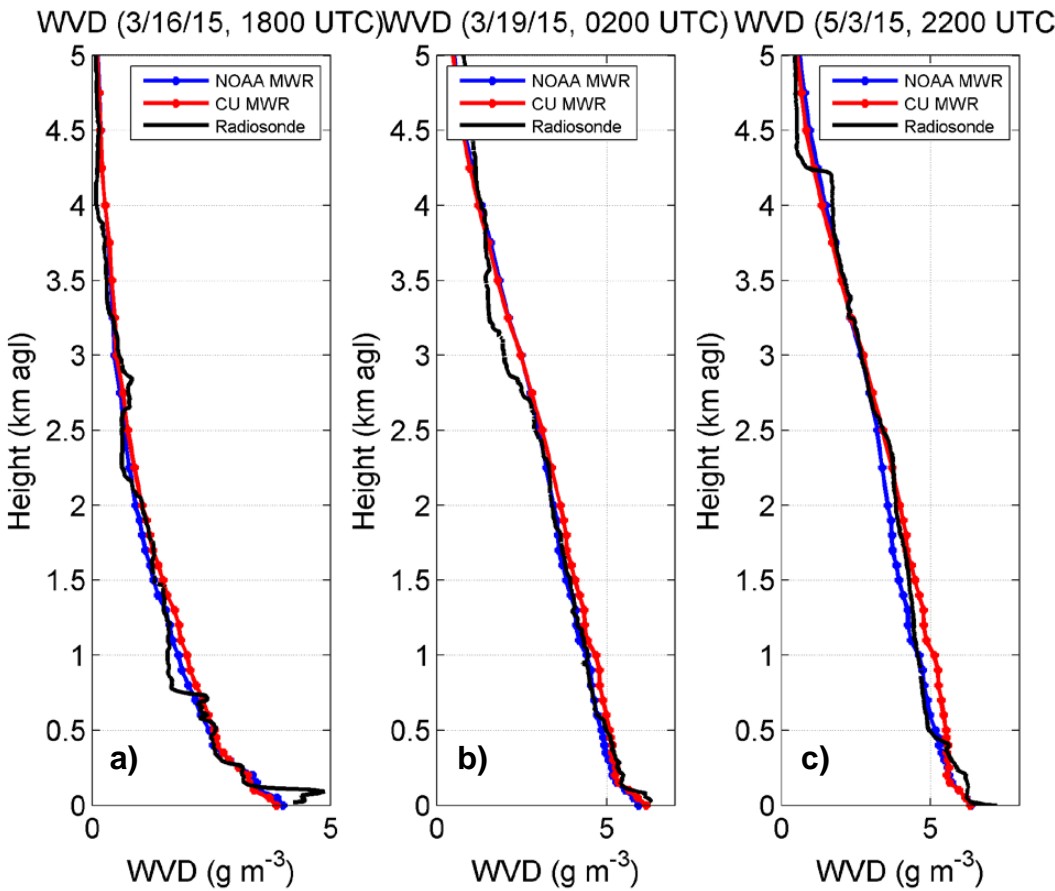

**Figure 19: Vertical profiles of *WVD* as observed by MWRs (blue line: NOAA MWR; red**
**line: CU MWR) and radiosonde (black line) at a) 1800 LT (0000 UTC) on 16 March, b)**
**0200 LT (0800 UTC) on 19 March, and c) 2200 LT (0400 UTC) on 3 May 2015.**





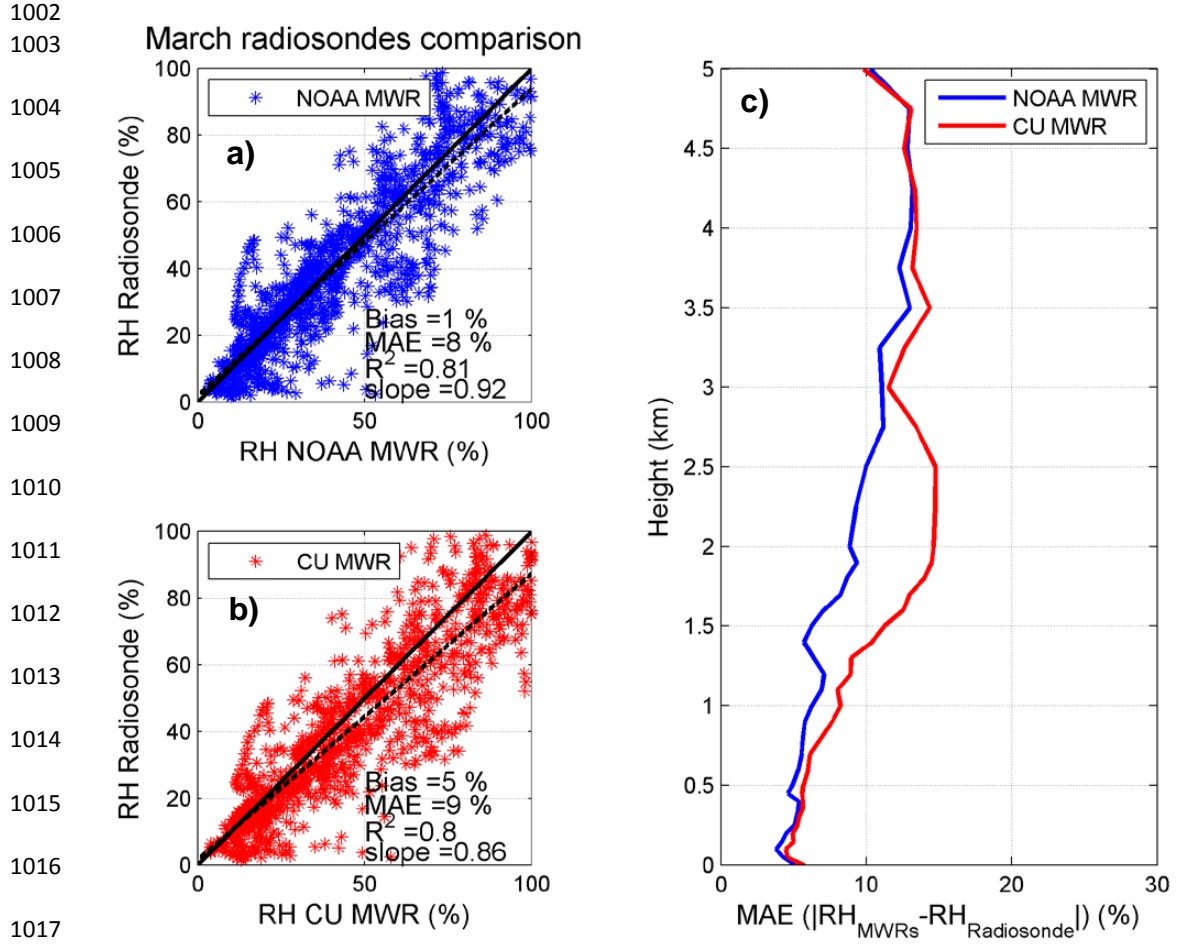

**Figure 20:** **MWR vs radiosonde comparison of *RH* over the March period of 38 radiosonde launches. a)-b) are one-to-one comparisons of *RH* observed by the radiosondes and the a) NOAA and b) CU MWR between the surface and 5 km AGL. One-on-one line is indicated as solid black line and the regression as dashed black line. c) Vertical profiles of MAE for the same variable.**





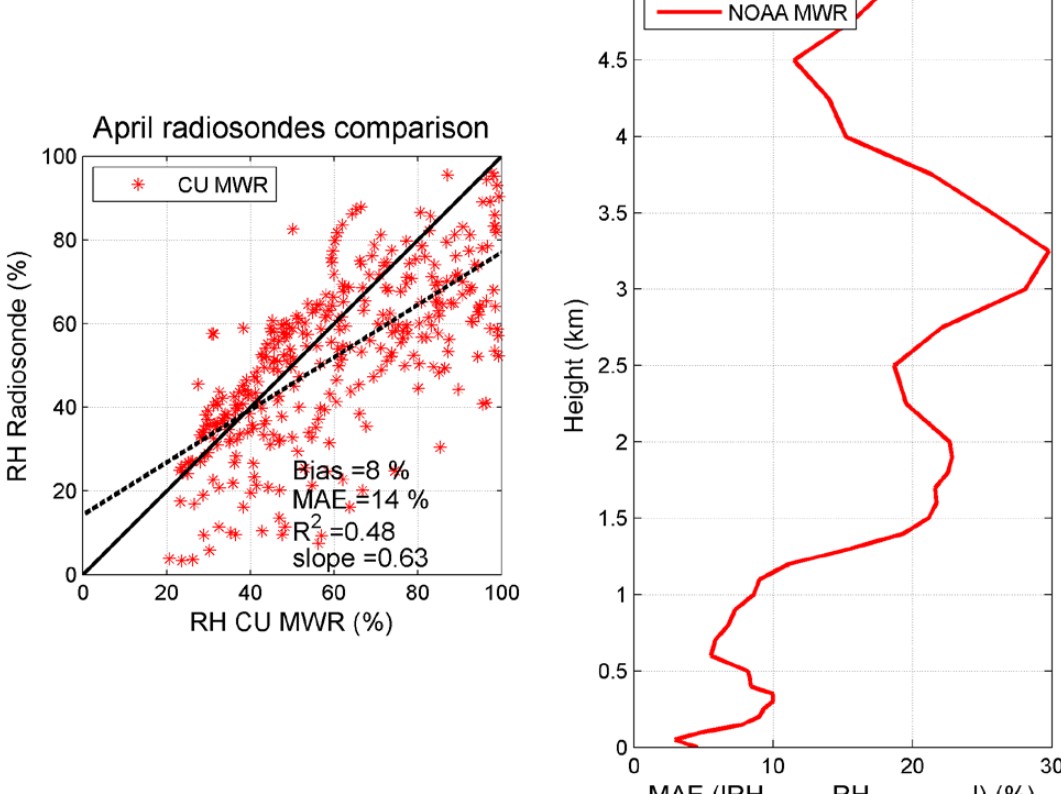

1025

**Figure 21:  Same as in Fig. 20, but for 15 and 20 – 22 April including 10 radiosonde**

**launches. Note that the pressure sensor of the NOAA MWR was broken between 5 – 27**

**April, therefore the NOAA MWR vs radiosonde comparison (*RH*) over this period is not**

**presented.**
























**Figure 22: Same as in Fig. 20, but for the May period of 13 radiosonde launches.**
























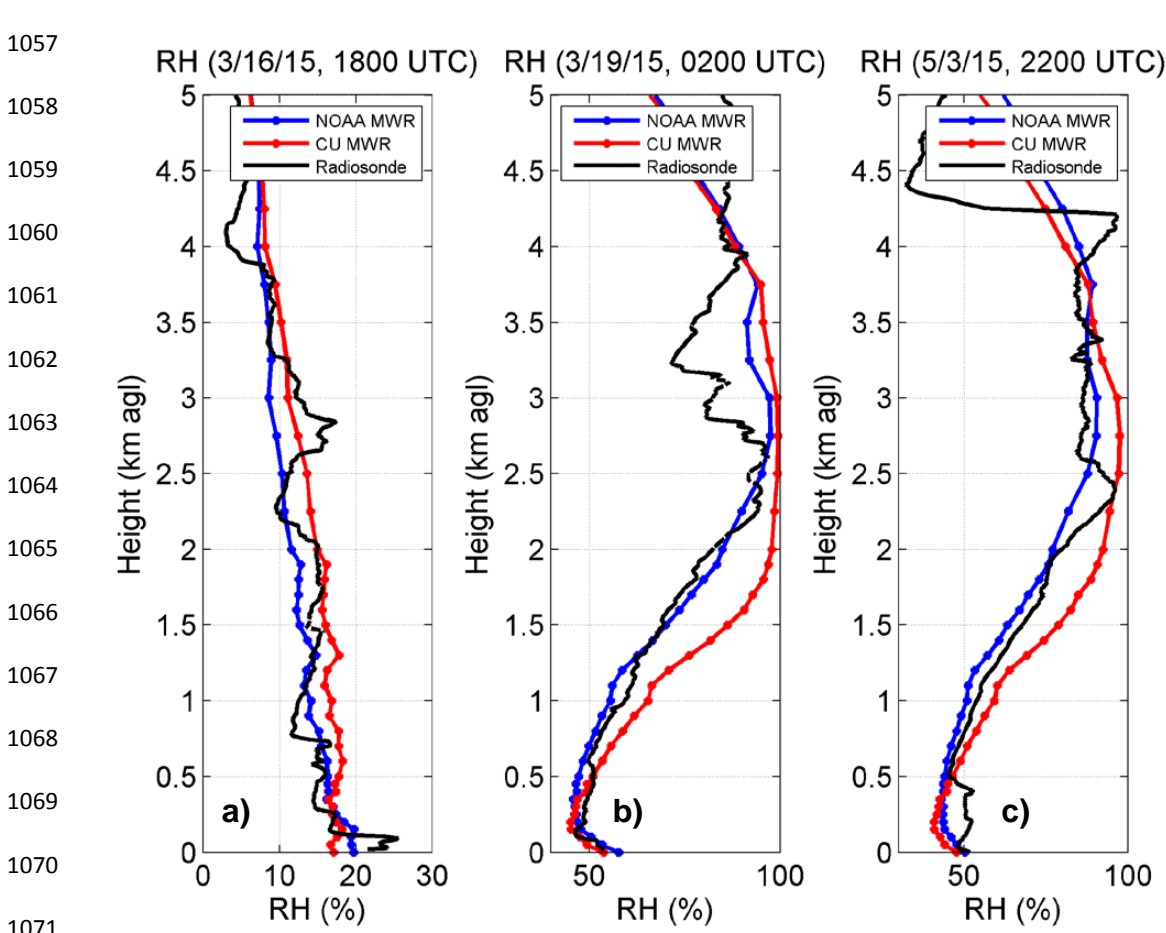

**Figure 23: Vertical profiles of _RH_ as observed by MWRs (blue line: NOAA MWR; red**
**line: CU MWR) and radiosonde (black line) at a) 1800 LT (0000 UTC) on 16 March, b)**
**0200 LT (0800 UTC) on 19 March, and c) 2200 LT (0400 UTC) on 3 May 2015.**


