# Peer review of "Assessing the accuracy of microwave radiometers and"

_Atmospheric Measurement Techniques, 2016_

## Referee Comment (RC1) · Anonymous Referee #1 · 24 Nov 2016

The manuscript AMT-2016-321 by Bianco et al. evaluates the accuracies of two MWRs and two RASSs with radiosonde soundings and 300-m meteorolgical tower observations based on the XPIA campaign data set. The authors show us the accuracy differences of temperature profiles of two identical MWRs and two different RASSs, which can benefit our better understanding on the measurement abilities of these instruments, especially on the random error between two identical MWRs. Another interesting point is the manuscript also evaluates the abilites of MWR and RASS for measuring temperature lapse rate, and the results may do good for wind energy applications. Overall, the manuscript is within the scope of the journal and it meets the scientific quality for AMT. Minor revisions should be considered by the authors before

the manuscript gets accepted for publication on AMT.

Minor comments:

(1) p13, line261, the "that" in " . . . with a slightly lower MAE that the CU MWR" should be "than".

(2) p13, line264-266, as shown in Fig. 3d, the temperature bias near the surface shows a negative value for NOAA MWR but a positive value for CU MWR, what's the explanation?

(3) p13, line267, the "if" in " . . . (an example if" should be "of".

(4) p13, line271, the "if" in " . . . an example if which is shown. . ." should be "of".

(5) p19, line397-404, the temperature MAE shows a smaller value in unstable conditions compared to stable conditions, could the authors give an explanation or discussion on it?

(6) the authors should check typing errors carefully.

---

## Referee Comment (RC2) · Anonymous Referee #2 · 21 Dec 2016

Review for **"Assessing the accuracy of microwave radiometers and radio acoustic sounding systems for wind energy applications"** by Laura Bianco et al. submitted to Atmospheric Measurement Techniques

**General comments:**

The manuscript presents results from a study of atmospheric boundary layer observations with remote sensing instruments. The focus was on the performance of microwave radiometers and wind profilers for temperature and humidity profiles. These observations were validated by frequent radiosondes and a 300-m high tower. The main findings are well presented and illustrated.

However, the discussion on the observed deviations between the different methods is quite short and incomplete, especially with focus on the microwave radiometers. The observed biases are in the range of calibration and retrieval uncertainties and should be therefore more thoroughly discussed. My main concern is that the observed differences between the two MWR profilers are caused by calibration offsets. To me, this result is not satisfying, and therefore, I suggest this manuscript to be published only after major revision.

The number of figures should also be reduced; the information gain is not very high when having all datasets separated into different months.

**Specific comments:**

p.8, line 158ff.: Which type of retrieval algorithm was used for the microwave radiometer profiles? You mention here "using historic radiosondes and a regression method or neural network". Please specify which of the two methods has been used? Regression or neural network?

Do you apply a separate algorithm for relative humidity, or do you calculate relative humidity from temperature and water vapor density? Please specify!

The accuracy of microwave profiles in the presence of rain does also deteriorate because of scattering of radiation due to raindrops in the atmosphere, not only due to a wet radome (p.9, line 168). Please clarify!

p. 9, lines 172: the resolution of the profiles (50/100/250 m) is not the "true" resolution of the profile which would be the number of independent layers you can retrieve. Some information on that is given in Löhnert and Maier (2012, AMT) or Crewell and Löhnert (2007, IEEE TGRS). There also uncertainties are given for temperature profiles from microwave radiometers in the lower troposphere. Please also discuss your results in the light of the findings of these two articles mentioned here.

p. 9, lines 176-179: why can't you take the pressure value from the other radiometer and do post-processing of the raw data? With that you could extend your dataset very easily!

p. 11, lines 226ff: To me, this mean absolute error can only be explained by inaccurate calibration of at least one of the instruments, considering the fact that you used the same retrieval algorithm. This could be assessed if the observed brightness temperatures from the MWRs are compared to each other. Another possibility for the inconsistency between the two MWRs could be inaccurate ground pressure or temperature sensors, since these data are included in the retrieval algorithm.

p. 12, line 249: What do you mean by "temperature saturation"? Under an angle of 15°, the atmosphere becomes saturated also for frequencies that are not saturated when zenith looking.

p.12, line 250: Friedrich et al., 2012 do not talk about off-zenith microwave observations.

p.13, lines 270ff: The accuracy might also decrease because of elevated temperature inversions that cannot be detected by the MWR

p.15, lines 312-313: The observable range for wind profilers decreases with increasing frequency, this is a well-known feature.

p. 16, lines 325ff: Again, the bias is most likely caused by calibration uncertainties or application of a not suitable retrieval algorithm.

p.20, lines 428 ff.: The uncertainties in relative humidity are higher due to both the temperature and the water vapor uncertainty that influence this variable.

**Technical corrections:**

Throughout the manuscript, you use the unit °C for biases, errors and gradients. However, for comparison please use Kelvin "K", since this is an absolute unit. °C is only accepted for actual temperature values. Please change also the figure captures!

p.8, line 156: GHz (not GZ)

p.13, line 271: check phrase in brackets!

p.19, line 409: "10 m AGL" – this doesn't make any sense. Do you mean "1000 m AGL" ?

---

## Author Comment (AC1) · 9 Feb 2017

1) The manuscript AMT-2016-321 by Bianco et al. evaluates the accuracies of two MWRs and two RASSs with radiosonde soundings and 300-m meteorological tower observations based on the XPIA campaign data set. The authors show us the accuracy differences of temperature profiles of two identical MWRs and two different RASSs, which can benefit our better understanding on the measurement abilities of these instruments, especially on the random error between two identical MWRs. Another interesting point is the manuscript also evaluates the abilities of MWR and RASS for measuring temperature lapse rate, and the results may do good for wind energy

applications. Overall, the manuscript is within the scope of the journal and it meets the scientific quality for AMT. Minor revisions should be considered by the authors before the manuscript gets accepted for publication on AMT.

2) Answer: We thank the Reviewer for these encouraging comments, as well as for the constructive suggestions. We have modified the manuscript according to the Reviewer's suggestions and hope to have addressed his/her concerns in the revised version of the manuscript.

Minor comments:

1) p13, line261, the "that" in ". . . with a slightly lower MAE that the CU MWR" should be "than".

2) Answer: We thank the Referee for catching the typo.

3) Changes: The text was modified in the revised version of the manuscript as suggested.

1) p13, line264-266, as shown in Fig. 3d, the temperature bias near the surface shows a negative value for NOAA MWR but a positive value for CU MWR, what's the explanation?

2) Answer: We thank the Referee for suggesting us to investigate into this. The neural net retrieval algorithm uses ground-based observations of temperature and pressure to derive the vertical temperature profile. The bias close to the surface is most likely related to the differences in temperature and pressure from the surface observations. We observed small variations in temperature throughout the day, which might be related to boundary layer heating. We also observed that differences in pressure had a great impact on the accuracy of the temperature. We did look into the accuracy of the surface sensors comparing to the 2m measurements collected at the base of the 300-m tower. For the pressure we found that:

- For the period of analysis between Mar 9 and Apr 4 the NOAA MWR surface sensor

had a bias in pressure (p_NOAA_MWR -p_TOWER) equal to -6 mb, while the CU MWR had a bias in pressure (p_CU_MWR -p_TOWER) equal to 0 mb.

- For the period of analysis between Apr 29 and May 7 the NOAA MWR surface sensor had a bias in pressure (p_NOAA_MWR -p_TOWER) equal to -1 mb, while the CU MWR had a bias in pressure (p_CU_MWR -p_TOWER) equal to 0 mb.

We note that the failure of the surface pressure sensor between 5-27 April produced large differences in the retrieved profiles, so large that we had to avoid including that period of NOAA MWR data into the analysis. For this reason, we can expect that the differences of 6 mb between the NOAA and CU MWRs surface sensors might be the cause of the opposite biases found at the lowest levels of Fig. 3d, while those of Fig. 5d (after the surface sensor was replaces) are of the same sign.

3) Changes: Some text was added to the revised version of the manuscript about this on page 14: "For the March comparison, we note that biases are opposite for the NOAA MWR and CU MWR (Fig. 3d). Since surface observations of temperature and pressure are important for the retrieval algorithm, we analyzed surface observations of the two MWRs. Differences in surface pressures between the two MWRs and a surface met station on the order of ∼6 mb were observed for the March period and only ∼1 mb for the May period. Note that the NOAA MWR surface sensor was not functioning between 5 – 27 April. We believe that the differences of ∼6 mb between the NOAA and CU MWRs surface sensors are the likely the cause of the opposite biases found at the lowest levels of Fig. 3d, while those of Fig 5d, after the surface sensor was replaced and the differences were only ∼1 mb, are of the same sign."

1) p13, line267, the "if" in "... (an example if" should be "of".

2) Answer: We thank the Referee for catching the typo.

3) Changes: The text was modified in the revised version of the manuscript as suggested.

1) p13, line271, the "if" in "... an example if which is shown..." should be "of".

2) Answer: We thank the Referee for catching the typo.

2) Changes: The text was modified in the revised version of the manuscript as suggested.

1) p19, line397-404, the temperature MAE shows a smaller value in unstable conditions compared to stable conditions, could the authors give an explanation or discussion on it?

2) Answer: This is an interesting question. Most of the cases captured in Fig. 14b (stable) are cases with a surface inversion in the morning. When we analyzed individual cases, we can see that the MWR's sometimes have difficulties correctly representing the depth and the slope of the temperature inversion which is probably due to the coarser resolution (compared to soundings or tower observations). The unstable cases are usually profiles with a straight temperature line taken throughout the day. Even though the temperature decreases with height, most of the cases are just slightly unstable. As a result, we would expect the "stable" profiles to have a higher MAE due to the uncertainties in depth and slope of the surface inversion.

3) Changes: Included explanation in the text: "Note that stable conditions might generate profiles with an inversion at the surface. Smaller MAE's occurred in unstable conditions (MAE = 0.8 K; Fig. 14a) compared to stable conditions (MAE = 1.2 K; Fig. 14b). Larger MAE values in stable conditions might indicate that the MWR has difficulties accurately capturing the depth and the slope of the surface inversion."

1) The authors should check typing errors carefully.

2) Answer: We thank the Referee for the suggestion.

3) Changes: The manuscript was checked for typos and we hope we found them all.
* * *

---

## Author Comment (AC2) · 9 Feb 2017

Review for "Assessing the accuracy of microwave radiometers and radio acoustic sounding systems for wind energy applications" by Laura Bianco et al. submitted to Atmospheric Measurement Techniques

General comments:

1) The manuscript presents results from a study of atmospheric boundary layer observations with remote sensing instruments. The focus was on the performance of microwave radiometers and wind profilers for temperature and humidity profiles. These observations were validated by frequent radiosondes and a 300-m high tower. The

main findings are well presented and illustrated.

2) Answer: We thank the Reviewer for these encouraging comments, as well as for the constructive suggestions. We have modified the manuscript according to the Reviewer's suggestions and hope to have addressed his/her concerns in the revised version of the manuscript.

1) However, the discussion on the observed deviations between the different methods is quite short and incomplete, especially with focus on the microwave radiometers. The observed biases are in the range of calibration and retrieval uncertainties and should be therefore more thoroughly discussed. My main concern is that the observed differences between the two MWR profilers are caused by calibration offsets. To me, this result is not satisfying, and therefore, I suggest this manuscript to be published only after major revision.

2) Answer: Please see answers to this point in the "Specific comments" section below.

1) The number of figures should also be reduced; the information gain is not very high when having all datasets separated into different months.

2) Answer: Since the surface sensor of one of the MWRs was replaced half way through the campaign we prefer to keep the different periods separate in the MWR analysis. Also, one objective of this paper is to determine the performance of the different instruments under different weather conditions, which is unique and useful in terms of instrument performance for individual seasons. As such, the authors still believe that separating the observations in different periods is of great value to the reader.

3) Changes: We revised the figures and removed figures 17 and 21, although the discussion on the results is still included in the text. Text in the revised version of the manuscript has been modified to accommodate for the different figure numbers.

Specific comments:

1) p.8, line 158ff.: Which type of retrieval algorithm was used for the microwave radiometer profiles? You mention here "using historic radiosondes and a regression method or neural network". Please specify which of the two methods has been used? Regression or neural network?

2) Answer: We apologize for the confusion. The retrieved profiles are retrieved using neural networks that are trained on 5 years of site-specific radiosonde data.

3) Changes: The text was modified in the revised version of the manuscript from: "using historic radiosondes and a regression methods or neural network" to "using distinct neural networks that are trained on 5-years of site-specific radiosonde data"

1) Do you apply a separate algorithm for relative humidity, or do you calculate relative humidity from temperature and water vapor density? Please specify!

2) Answer: The relative humidity profiles are retrieved using a separate RH neural network trained on radiosonde data. The brightness temperatures are used as inputs into the RH neural net, but the profiles are retrieved independently of temperature and water vapor density, not calculated.

3) Changes: The text was modified in the revised version of the manuscript as mentioned in the answer above, specifying that the neural networks are "distinct".

1) The accuracy of microwave profiles in the presence of rain does also deteriorate because of scattering of radiation due to raindrops in the atmosphere, not only due to a wet radome (p.9, line 168). Please clarify!

2) Answer: We thank the Referee for the clarification.

3) Changes: The text was modified in the revised version of the manuscript from: "these instruments become less accurate in the presence of rain as some water deposits on the radome", to: "These instruments become less accurate in the presence of rain because of scattering of radiation due to raindrops in the atmosphere; also, although the instruments use a hydrophobic radome and forced airflow over the surface of the radome during rain, some water can still deposits on the radome", following the

suggestion of the Referee.

1) p. 9, lines 172: the resolution of the profiles (50/100/250 m) is not the "true" resolution of the profile which would be the number of independent layers you can retrieve. Some information on that is given in Löhnert and Maier (2012, AMT) or Crewell and Löhnert (2007, IEEE TGRS). There also uncertainties are given for temperature profiles from microwave radiometers in the lower troposphere. Please also discuss your results in the light of the findings of these two articles mentioned here.

2) Answer: We thank the Referee for the suggestion.

3) Changes: Regarding the "resolution of the profiles" we agree that this is not a "true" resolution, so we modified the text in the revised version of the manuscript from: "The vertical resolution of the retrieved profiles ranged from 50 m between the surface and 0.5 km AGL; 100 m between 0.5 to 2 km AGL; and 250 m between 2 and 10 km AGL." to "The nominal vertical levels of the retrieved profiles ranged from 50 m between the surface and 0.5 km AGL; 100 m between 0.5 to 2 km AGL; and 250 m between 2 and 10 km AGL. Note that we will refer to these levels as vertical resolution throughout the manuscript.". The references to Crewell and Löhnert (2007) and Löhnert and Maier (2012) were added to the introduction.

1) p. 9, lines 176-179: why can't you take the pressure value from the other radiometer and do post-processing of the raw data? With that you could extend your dataset very easily!

2) Answer: We agree with the Referee that we could have used the pressure values from the CU MWR surface sensor and reprocess the NOAA MWR dataset to retrieve new profiles, but this would have been completely in opposition to one of the areas we are trying to investigate in this manuscript: to look at the random differences between two identical MWRs, which are at the current time used in an one-year field campaign in Oregon and Washington state (USA). As we discussed (see answer below), these differences can be attributed to differences in the calibration, as well as differences in

the accuracy of the surface sensor values, which are included in the retrieval of the profiles. If we use the surface values from one MWR into the other one, than this point is going to be completely missed.

3) Changes: We didn't change the way we performed our analysis.

1) p. 11, lines 226ff: To me, this mean absolute error can only be explained by inaccurate calibration of at least one of the instruments, considering the fact that you used the same retrieval algorithm. This could be assessed if the observed brightness temperatures from the MWRs are compared to each other. Another possibility for the inconsistency between the two MWRs could be inaccurate ground pressure or temperature sensors, since these data are included in the retrieval algorithm.

2) Answer: We thank the Referee for suggesting us to investigate into this. Both MWRs were calibrated using an external liquid nitrogen target and an internal ambient target performed at the factory (Radiometrics Corporation) and for this reason we expect them to have received the optimal possible calibration. Nevertheless, we looked at the differences in the brightness temperature from the MWRs, as suggested by the Referee. We compared brightness temperatures of the two MWRs for each retrieved channel for one day finding almost all channels in pretty good agreement (<2 K difference) with some agreeing exactly. However we agree that even a slight difference is enough to skew the neural network retrievals slightly. This would contribute to the explanation of the differences between the two instruments. Some text was added to the revised version of the manuscript discussing these findings (page 8). Also, we did look into the accuracy of the surface sensors comparing to the 2m measurements collected at the base of the 300-m tower. For the pressure we found that:

- For the period of analysis between Mar 9 and Apr 4 the NOAA MWR surface sensor had a bias in pressure (p_NOAA_MWR -p_TOWER) equal to -6 mb, while the CU MWR had a bias in pressure (p_CU_MWR -p_TOWER) equal to 0 mb.

- For the period of analysis between Apr 29 and May 7 the NOAA MWR surface sensor

had a bias in pressure (p_NOAA_MWR -p_TOWER) equal to -1 mb, while the CU MWR had a bias in pressure (p_CU_MWR -p_TOWER) equal to 0 mb.

We note that the failure of the surface pressure sensor between 5-27 April produced large differences in the retrieved profiles, so large that we had to avoid including that period of NOAA MWR data into the analysis. For this reason we can expect that the differences of 6 mb between the NOAA and CU MWRs surface sensors might be the cause of the opposite biases found at the lowest levels of Fig. 3d, while those of Fig. 5d (after the surface sensor was replaces) are of the same sign.

3) Some text was added to the revised version of the manuscript discussing these findings (page 14): "For the March comparison, we note that biases are opposite for the NOAA MWR and CU MWR (Fig. 3d). Since surface observations of temperature and pressure are important for the retrieval algorithm, we analyzed surface observations of the two MWRs. Differences in surface pressures between the two MWRs and a surface met station on the order of ∼6 mb were observed for the March period and only ∼1 mb for the May period. Note that the NOAA MWR surface sensor was not functioning between 5 – 27 April. We believe that the differences of ∼6 mb between the NOAA and CU MWRs surface sensors are the likely the cause of the opposite biases found at the lowest levels of Fig. 3d, while those of Fig 5d, after the surface sensor was replaced and the differences were only ∼1 mb, are of the same sign."

1) p. 12, line 249: What do you mean by "temperature saturation"? Under an angle of 15°, the atmosphere becomes saturated also for frequencies that are not saturated when zenith looking.

2) Answer: This sentence has been removed.

1) p.12, line 250: Friedrich et al., 2012 do not talk about off-zenith microwave observations.

2) Answer: Friedrich et al. (2012) actually uses the average temperatures

from the 15deg off-zenith scans as published in the supporting information (grl28883-sup-0002-txts01.pdf, page 3 lines 50-51): "For the comparison experiment, the temperature measured by the radiometer was averaged between the $15°$ elevation scan pointing north and the $15°$ elevation scan pointing south." (http://onlinelibrary.wiley.com/store/10.1029/2011GL050413/asset/supinfo/grl28883-sup-0002-txts01.pdf?v=1&s=a5198cad98c05e19a5cf7d5e5344953fa306065b)

3) No Changes: changes have been applied to the text.

1) p.13, lines 270ff: The accuracy might also decrease because of elevated temperature inversions that cannot be detected by the MWR

2) Answer: We thank the Referee for the suggestion.

3) Changes: The text was modified in the revised version of the manuscript from: "Above 1.5 km, for some of the profiles, radiosonde temperatures strongly differed from the MWR observations (an example of which is shown in Fig. 6b), which might be related to strong observed winds aloft (winds larger than 10 m s-1 for these circumstances, not shown) that transported the sounding farther away from the MWR encountering different air masses", to: "Above 1.5 km, for some of the profiles, radiosonde temperatures strongly differed from the MWR observations (an example of which is shown in Fig. 6b), which might be related to the presence of elevated temperature inversions that cannot be detected by the MWR or to strong observed winds aloft (winds larger than 10 m s-1 for these circumstances, not shown) that transported the sounding farther away from the MWR encountering different air masses".

1) p.15, lines 312-313: The observable range for wind profilers decreases with increasing frequency, this is a well-known feature.

2) Answer: We thank the Referee for the reminder. We definitely didn't want to claim that we are discovering something new about the different height coverages of the two RASS systems (915-MHz and 449-MHz), but we wanted to point out that because

of this even more caution should be engaged in the use of the virtual temperature corrected for the vertical velocity for the 915-MHz RASS as this correction is even more detrimental compared to the 449-MHz RASS.

3) Changes: The text was modified in the revised version of the manuscript from: "Moreover, the correction is more negative on the 915-MHz RASS Tv which is an indication that the vertical velocity measurements are more difficult for this system compared to the 449-MHz RASS." To: "Moreover, the correction is more negative on the 915-MHz RASS Tv which is in agreements with the fact that the vertical velocity measurements are less accurate for this system compared to the 449-MHz RASS (Ecklund et al., 1990)." We also added the Reference: "Ecklund, W. L., D. A. Carter, B. B. Balsley, P. E. Currier, J. L. Green, B. L. Weber, K. S. Gage, 1990: Field tests of a lower tropospheric wind profiler, Radio Science, 25, 899-906. doi:10.1029/RS025i005p00899" to the list of References.

1) p. 16, lines 325ff: Again, the bias is most likely caused by calibration uncertainties or application of a not suitable retrieval algorithm.

2) Answer: We think this might be related to the accuracy in the surface sensor and we included a sentence about it in the revised version of the manuscript.

3) Changes: Some text was included in the revised version of the manuscript discussing the importance of the accuracy of surface sensor values.

1) p.20, lines 428 ff.: The uncertainties in relative humidity are higher due to both the temperature and the water vapor uncertainty that influence this variable.

2) Answer: As mentioned above, the relative humidity profiles are retrieved using a separate RH neural network trained on radiosonde data, not calculated.

3) Changes: The text was modified in the revised version of the manuscript as mentioned above.

Technical corrections:

1) Throughout the manuscript, you use the unit °C for biases, errors and gradients. However, for comparison please use Kelvin "K", since this is an absolute unit. °C is only accepted for actual temperature values. Please change also the figure captures!

2) Answer: We thank the Referee for the suggestion.

3) Changes: The text was modified through the revised version of the manuscript to use "K" instead of "°C" when referring to MAEs, Biases, and error gradients, as suggested by the Referee. Also, all the legends, labels, and text in the tables were modified to reflect this change. For this reason figures 2, 3, 4, 5, 7, 8, 9, 10, 11, 12, and 14 had to be revised accordingly.

1) p.8, line 156: GHz (not GZ)

2) Answer: We thank the Referee for catching the typo.

3) Changes: The text was modified in the revised version of the manuscript as suggested.

1) p.13, line 271: check phrase in brackets!

2) Answer: We thank the Referee for catching the typo.

3) Changes: The text was modified in the revised version of the manuscript to: "(an example of which is shown in Fig. 6b)".

1) p.19, line 409: "10 m AGL" – this doesn't make any sense. Do you mean "1000 m AGL"?

2) Answer: We thank the Referee for catching the typo.

3) Changes: The text was modified in the revised version of the manuscript from: "10 m AGL" to: "10 km AGL", we apologize for the confusing typo.